# Energy-Based Flow Matching for Generating 3D Molecular Structure

**Wenyin Zhou** [1] **Christopher Iliffe Sprague** [1 2 3] **Vsevolod Viliuga** [2 4 5] **Matteo Tadiello** [2 4] **Arne Elofsson** [2 4]
**Hossein Azizpour** [1 2]

## Abstract

Molecular structure generation is a fundamental problem that involves determining the 3D positions of molecules' constituents. It has crucial biological applications, such as molecular docking, protein folding, and molecular design. Recent advances in generative modeling, such as diffusion models and flow matching, have made great progress on these tasks by modeling molecular conformations as a distribution. In this work, we focus on flow matching and adopt an energy-based perspective to improve training and inference of structure generation models. Our view results in a mapping function, represented by a deep network, that is directly learned to *iteratively* map random configurations, i.e. samples from the source distribution, to target structures, i.e. points in the data manifold. This yields a conceptually simple and empirically effective flow matching setup that is theoretically justified and has interesting connections to fundamental properties such as idempotency and stability, as well as the empirically useful techniques such as structure refinement in AlphaFold. Experiments on protein docking as well as protein backbone generation consistently demonstrate the method's effectiveness, where it outperforms recent baselines of task-associated flow matching and diffusion models, using a similar computational budget.

## 1. Introduction

Structure prediction, the task to determine a molecule's 3D structure, is a fundamental problem in structural biology for understanding diverse biological mechanisms (Kessel & Ben-Tal, 2018). It finds various applications, such as protein docking, which generates the bound structure of protein-ligand complex, and *de novo* protein design, which creates novel proteins fulfilling certain desirable biological functions. Regression-based structure prediction models (Jumper et al., 2021; Baek et al., 2021; Lin et al., 2023) have demonstrated remarkable performance by using a deep neural network to make a point estimate of the structure. However, these methods do not capture aleatoric uncertainty due to the possibility of multiple molecular conformations.

In contrast to regression-based models, generative methods, such as diffusion (Song & Ermon, 2019; Ho et al., 2020; Song et al., 2021) and flow matching models (Lipman et al., 2023; Albergo & Vanden-Eijnden, 2023; Liu et al., 2022), can generate several possible configurations modeling the positional distribution of molecules. Owing to this advantage, sampling-based approach has recently attracted huge interest in molecular structure generation, including molecular conformer generation (Hoogeboom et al., 2022), molecular docking (Corso et al., 2023), and protein design (Watson et al., 2023; Campbell et al., 2024).

Here, importantly, we first observe that these families of generative modeling are generally reminiscent of the computational models for molecular structure determination. Particularly, in computational models, a stable equilibrium structure is sought as the local minimum of an energy function governed by the Boltzmann distribution $p(x) \propto \exp(-E(x))$, which resembles modeling (and generating from) the data distribution by estimating the score – or a corresponding energy – of its density. In this work, we make this connection explicit and propose a simple modification of the objective in flow matching that directly minimizes an energy function.

Specifically, we take the reconstruction error $\|x - \hat{x}\|$ as the energy function and locally construct the energy landscape using contrastive predictions sampled from the flow model. Leveraging a data parameterization (Stärk et al., 2023; Yim et al., 2023a) of flow matching, minimizing our devised energy, quite intriguingly, translates into training a neural network that iteratively predicts and refines the generated sample as shown in Fig. 1, rendering the neural network, an approximated *idempotent* function.

---

[1]KTH Royal Institute of Technology, Stockholm, Sweden [2]Science for Life Laboratory, Sweden [3]The Alan Turing Institute, London, United Kingdom [4]DBB at Stockholm University, Sweden [5]Max Planck Institute for Polymer Research, Mainz, Germany. Correspondence to: Wenyin Zhou <wenyinz@kth.se>.

*Proceedings of the 42$^{nd}$ International Conference on Machine Learning*, Vancouver, Canada. PMLR 267, 2025. Copyright 2025 by the author(s).

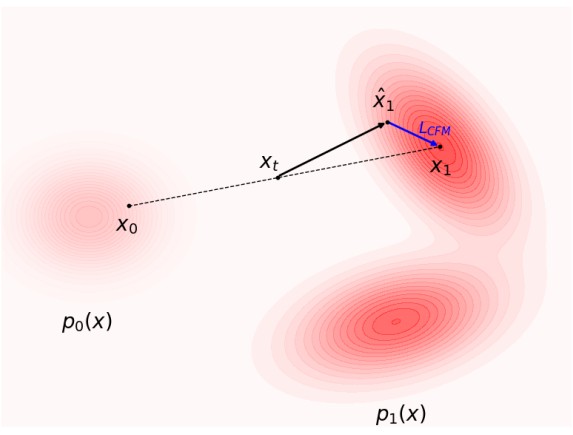

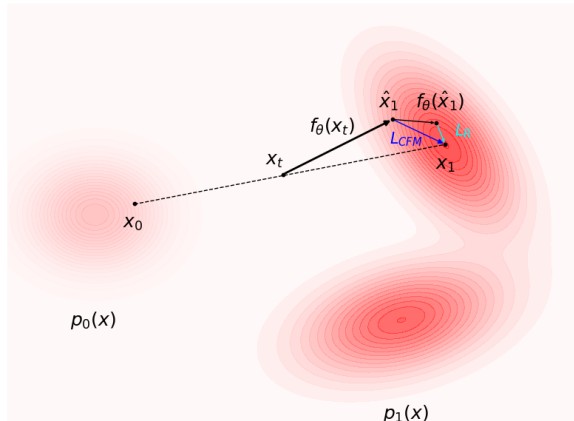

(a) Flow matching data parameterization.

(b) Flow Matching: idempotent flow map

*Figure 1.* **Training paradigm of standard flow matching and IDFlow.** $x_0$ and $x_1$ are samples from the source and target distribution $p_0(x)$ and $p_1(x)$, $x_t$ is the linear interpolant between the source and target sample, $\hat{x}_1$ is the prediction of the network, $f_\theta(\hat{x}_1)$ is the refined prediction, $L_{\text{CFM}}$ is the conditional flow matching loss and $L_R$ is the refinement loss. (a) Flow matching: directly predict the data $\hat{x}_1$ given the interpolant $x_t$. (b) IDFlow: a mapping function represented by a deep neural network $f_\theta$, iteratively predicts and refines the prediction. The energy is estimated to be the input output structure difference after refinement.

Notably, our trained idempotent function (IDFlow) simultaneously serves as both a neural sampler and a refiner, leading to the characterization of a predictor-refiner sampler, analogous to the predictor-corrector sampler in diffusion models (Song et al., 2021). This approach iteratively refines the sampling trajectory toward points where the energy function approximates zero, effectively enhancing flow matching as an energy-based model but crucially without significant increase in training cost. The key contributions are as follows.

- We adopt an energy-based model perspective for flow matching on sampling-based molecular structure generation tasks which respects the energy minimization.

- We provide a simple instance of such framework with negligible computational burden, IDFlow, which iteratively predicts and refines the sample and, importantly, draws interesting connections to related approaches, such as structure refinement.

- We conduct extensive experiments based on two recent flow matching models tackling three structure prediction tasks of (single and multi-ligand) docking and protein design using four public datasets. Our results show that IDFlow improves the structure generation performance over the recent task-associated flow matching and diffusion models.

## 2. Background

### 2.1. Energy-Based Models

Energy-Based Models (EBMs) consider a positive energy function $E_\theta(x) \in \mathbb{R}_+$ assigning a scalar value to each data

point $x$. Learning within this paradigm requires devising an energy function and a loss function to shape the energy landscape. For instance, the $L_2$ regression loss could be seen as directly using the energy function for the loss and the exact format of the energy function (referred to as the **energy architecture** hereafter) is the $L_2$ norm between the network output $f_\theta(x)$ and the label $y$:

$$\textbf{Energy}: \quad E_\theta(x) = ||f_\theta(x) - y||^2 \quad (1)$$
$$\textbf{Loss}: \quad L_\theta(x) = E_\theta(x), \quad (2)$$

where $(x, y)$ is a training pair of input and annotation. Training with this strategy primarily reduces the energy of the training examples. In general, there are two ways of training the EBMs: 1) contrastive approaches which maximize the energy margin between positive and negative training examples, and 2) regularized methods that minimize the volume of low-energy space while avoiding the collapse of the overall energy landscape, e.g., regularizing the latent space to be close to the standard Gaussian as low energy space in variational autoencoders (Kingma & Welling, 2013). Particularly, the collapse occurs when the network produces identical outputs regardless of the input. One contrastive method of training EBMs is learning a function that maps the points off the data manifold to the points on the data manifold (LeCun, 2022). It is this latter approach that we adopt in this work to improve Flow Matching for structure prediction. For a detailed review of EBMs, see (LeCun et al., 2006).

## 2.2. Flow Matching

Flow matching (Lipman et al., 2023; 2024; Albergo & Vanden-Eijnden, 2023; Liu et al., 2022) is a simulation-free training method for continuous normalizing flows (Chen et al., 2018) that connects two arbitrary distributions with flexible design choices. This allows for building straight paths between any source and target sample, which has great potential for accelerated sampling. Specifically, flow matching aims to learn a time-dependent vector field $v_t(x)$ evolving the sample from the prior $p_0(x)$ to the target distribution $p_1(x)$ via an ordinary differential equation (ODE):

$$\frac{\mathrm{d}x}{\mathrm{d}t} = v_{\theta,t}(x), \tag{3}$$

where $x$ could either be defined over Euclidean space $\mathbb{R}^d$ or a Riemannian Manifold $\mathcal{M}$ (Chen & Lipman, 2024). To achieve this, the flow matching objective (Lipman et al., 2023) regresses the predicted vector field to the true vector field $u_t(x)$:

$$L_{\text{FM}}(\theta) := \mathbb{E}_{\substack{t \sim \mathcal{U}(0,1) \\ x \sim p_t(x)}} \| v_{\theta,t}(x) - u_t(x) \|_2^2. \tag{4}$$

However, this objective function cannot be leveraged for training as the true vector field is intractable in practice. Previous work (Lipman et al., 2023; Tong et al., 2024) shows that with the construction of the conditional probability paths, the conditional flow matching (CFM) objective shares the same parameter gradient as the flow matching objective $\nabla_\theta L_{\text{FM}} = \nabla_\theta L_{\text{CFM}}$, such that learning the conditional vector field would also optimize for the marginal vector field:

$$L_{\text{CFM}}(\theta) := \mathbb{E}_{\substack{t \sim \mathcal{U}(0,1) \\ x \sim p_t(x|x_0,x_1) \\ x_0 \sim p(x_0) \\ x_1 \sim q(x_1)}} \| v_{\theta,t}(x) - u_t(x \mid x_0, x_1) \|_2^2 \tag{5}$$

where $u_t(x|x_0,x_1)$ is the conditional vector field that generates the conditional probability path $p_t(x|x_0,x_1)$, and $x_0$ and $x_1$ are sampled from the prior (or source) distribution and the training (or target) set respectively. Notably, flow matching does not depend on any prior distribution (standard Gaussian in diffusion models) and allows flexible construction of probability paths. To reduce the transport cost, conditional optimal transport builds the probability path that linearly transfers the source sample to the target:

$$p_t(x|x_1,x_0) = \mathcal{N}(x|\mu_t(x), \sigma_t^2) \quad \mu_t(x) = tx_1 + (1-t)x_0 \tag{6}$$

where $\sigma_t$ is a predefined standard deviation which could be either constant or a time-dependent noise scheduler (e.g. $\sigma_t = \sqrt{t(1-t)}I_d$), and $\mu_t(x)$ is the mean of the conditional distribution.

$x_1$ **parameterization.** Similar to diffusion models (Watson et al., 2023), one can opt to parameterize the network output as either the vector field $u_t(x|x_0,x_1)$ or the data $x_1$ (Stärk et al., 2023; Jing et al., 2024). From the EBMs' perspective, the latter results in an objective function that directly pushes down the energy of the trajectory samples, which is desirable:

$$L_{\text{CFM}}(\theta) := \mathbb{E}\|f_{\theta,t}(x) - x_1\|_2^2. \tag{7}$$

The $x_1$ parameterization yields an Euler-like step sampling algorithm:

$$\frac{\mathrm{d}x}{\mathrm{d}t} = \frac{f_{\theta,t}(x) - x_t}{1 - t} \tag{8}$$

where $f_\theta$ is the mapping between a point from the sampling trajectory to the corresponding target data point, and is, hereafter, referred to as the *flow map*. The general format of the vector field can be found in the App. B.3. A more intuitive understanding of $f_\theta$ is the "**denoiser**" that approximates the expectation of $p_{1|t}(x_1|x)$:

$$\mathbb{E}[X_1|X_t = x] \approx f_{\theta,t}(x). \tag{9}$$

In this work, we focus on the $x_1$ parameterization for its connections to the EBMs loss and generative adversarial nets (Goodfellow et al., 2014).

**Riemannian flow matching.** Flow matching is extended to the Riemannian manifold by constructing a premetric that measures the proximity of a point $x$ to the target $x_1$ for defining the vector field (Chen & Lipman, 2024). The core idea is to define the premetric as the geodesic distance, such that the geodesic interpolant $x_t$ can be efficiently computed with the exponential and logarithmic map on simple manifolds (e.g. the N-D sphere) in closed-form:

$$\textbf{Euclidian:} \quad x_t = tx_1 + (1-t)x_0, \tag{10}$$

$$\textbf{Riemannian:} \quad x_t = \exp_{x_0}(t\log_{x_0}(x_1)). \tag{11}$$

Specifically, the exponential and logarithmic map of the manifold SO(3) can be calculated using the well-known Rodrigues' formula. This has been applied to generating protein backbones (Yim et al., 2023a; Bose et al., 2024; Huguet et al., 2024; Wagner et al., 2024) and robot motion learning (Braun et al., 2024).

## 3. Method

### 3.1. Task Formulation and Notation

We consider two widely adopted tasks, molecular docking and protein backbone generation for the investigation of the method on generating 3D molecular structure.

**Protein docking.** For the molecular docking task, we represent the protein structure using Cartesian coordinates as $y \in \mathbb{R}^{3 \times n_p}$ and the ligand structure as $x \in \mathbb{R}^{3 \times n_l}$, where $n_p$ and $n_l$ are the number of protein residues and ligand atoms,

respectively. The aim is to learn the conditional distribution $p(x|y)$, i.e. the docking distribution of the ligand $x$ given the protein structure $y$.

**Single ligand docking** assumes a unique protein-ligand pair $(x, y)$, such that the generative model learns the binding modes of the protein $y$ with *a single sample* from the conditional distribution $p(x|y)$.

**Multi-ligand docking** assumes multiple pairs of protein $y$ and ligands $x_i$: $(x_1, y), (x_2, y), ..., (x_n, y)$, where $n$ is the number of ligands for protein $y$. The training signal of multi-ligand docking can be enriched with several docking conformations $x_i$ provided for learning the conditional distribution $p(x|y)$, but also increases the learning difficulty. This task finds application in multi-ligand binding pocket design, such as the enzyme design for multiple reactants.

**Protein backbone generation** represents each protein residue by a frame representation $(r, s) \in \mathrm{SE}(3)$, a rotation $r \in \mathrm{SO}(3)$ and a translation $s \in \mathrm{T}(3)$. Hence, a protein $y$ is modeled by the $N$ compositions of $\mathrm{SE}(3)$ group, where $N$ is the number of protein residues. As such, the generative model is defined over the manifold $y \in \mathcal{M} \equiv \mathrm{SE}(3)^N$ for learning the protein distribution $p(y)$. The translation $s$ places each protein residue, particularly the $C_\alpha$ carbon, relative to the reference frame, and the rotation $r$ captures the local orientation of the residue. The detailed definition of the reference frame can be found in App. C.2.

### 3.2. Energy-Based Flow Matching

**Energy relaxation, confidence model and the EBMs.** Data-driven generated structures often tend to be suboptimal when evaluated against physiochemical energy functions. A common solution, known as energy relaxation, is to post-optimize the molecular poses with the energy function, so that the structure after postprocessing corresponds to the local energy minimum (Buttenschoen et al., 2024). In sampling-based molecular docking, a confidence model ranks the samples by outputting their corresponding confidence scores. In both cases, a scalar is produced for the molecular structure and can be collectively understood as the energy. To draw a connection, we can write down the energy of the generated sample $\hat{x}_1$ as:

$$E_\theta(\hat{x}_1) = D_\theta(\hat{x}_1), \qquad \hat{x}_1 = f_{\theta,t}(x), \qquad (12)$$

where $D_\theta$ is the energy architecture and $\hat{x}_1$ is the sample produced by the flow map $f_\theta$. For structure relaxation, the energy architecture $D_\theta$ is a specific biophysics-informed energy function, for example, the Amber energy (Duan et al., 2003), which involves the bond, angular, Lennard Jones, and Coulomb energy. Under the scenario of molecular docking, $D_\theta$ could be a separate neural network trained by generating samples for every training example and using the training annotations to produce binary labels to distinguish the posi-

tive (high-accuracy) and negative (low-accuracy) examples (Corso et al., 2023). Then the logits of such a network can be used as the confidence score (energy function) for an unseen example. In contrast to these two approaches, in this work, we propose to directly construct and learn an energy function within the Flow Matching model. However, from objective Eq. 7, it is evident that only the energy associated with the trajectory sample $x_t$ is pushed down, while the contrastive samples $\hat{x}_1$ generated by the flow model, unlike $x_1$, remain unaffected by this process. Therefore, the idea of the energy-based flow matching is precisely to better shape the energy landscape to improve training with contrastive samples $\hat{x}_1$ and to encourage reaching the minimum of the energy function as shown in Fig. 1.

**Shaping the loss landscape with $\hat{x}_1$.** We first need to define an energy function $E(\hat{x}_1) : \mathbb{R}^n \to \mathbb{R}_+$, such that high-probability data lies around its minima. Following (Zhao et al., 2017), a simple energy architecture could be the reconstruction error involving a certain function $G$:

$$E_\theta(\hat{x}_1) = ||G(\hat{x}_1) - \hat{x}_1||_2^2. \qquad (13)$$

Since the conversion from energy to probability could be achieved through the Boltzmann distribution, this energy function also aligns with the likelihood function in the context of conditional flow matching with the Gaussian assumption over the training example $x_1$, where the exponent is a quadratic function with a constant variance (App. B.4). Since the energy function shares a similar L2 form as the CFM loss, the loss landscape in the conformation space, which was originally only influenced by the observed $x_1(x_t$, as $t \to 1)$, now is further shaped by the estimated $\hat{x}_1$. Assuming $G$ is a function that perfectly maps any $\hat{x}_1$ to $x_1$ on the data manifold, Eq. 13 will assign high energy (larger reconstruction error) to 'bad' $\hat{x}_1$ and low energy (smaller reconstruction error) to 'good' $\hat{x}_1$. However, in principle, $G$ could be any neural network that keeps the dimensionality of the input data. Crucially, following this, during training, the flow map $f_{\theta,t}(x)$ receives the gradient not only from the CFM loss but also from the energy loss of Eq. 13 for the contrastive samples.

**Sample from the energy function.** With the energy function devised, we can define an energy-based density with the Boltzmann distribution:

$$p(x) = \frac{\exp(-E(x))}{Z}, \qquad (14)$$

where $Z$ is some unknown normalizing constant. Taking the derivative of the log likelihood with respect to $x$ yields:

$$\nabla_x \log(p(x)) = -\nabla_x E(x). \qquad (15)$$

Now, assume that we define a gradient flow vector field:

$$v(x) = -\nabla_x E(x). \qquad (16)$$

---

**Algorithm 1** Idempotent Flow Map Training

---

**Require:** prior distribution $p_0(x)$, data distribution $p_1(x)$
**while** Training **do**
  $x_0 \sim p_0(x_0), x_1 \sim p_1(x_1).$
  $t \sim U(0,1), m \sim U(0,1)$
  $\mu_t \leftarrow t \cdot x_1 + (1-t) \cdot x_0$
  $x \sim \mathcal{N}(\mu_t, \sigma_t^2 I)$
  **if** $m \leq 0.5$ **then**
    $k \sim \text{randint}(1, K_{\max})$
    With torch.no_grad():
      $\hat{x}_1 = f_{\theta,t}(x)$
    $x_1\_list = [\,]$
    **for** $i = 0, \ldots, k$ **do**
      $\hat{x}_1 \leftarrow f_{\theta,t}(\hat{x}_1.detach())$
      $x_1\_list.append(\hat{x}_1)$
    **end for**
    $L_{\text{R}} \leftarrow \frac{1}{|x_1\_list|} \sum_{\hat{x}_1 \in x_1\_list} \|\hat{x}_1 - x_1\|^2$
    $\theta \leftarrow \text{Update}(\theta, \nabla_\theta L_{\text{R}})$
  **else**
    $L_{\text{CFM}} \leftarrow \|f_{\theta,t}(x) - x_1\|^2$
    $\theta \leftarrow \text{Update}(\theta, \nabla_\theta L_{\text{CFM}})$
  **end if**
**end while**

---

**Algorithm 2** Predictor Refiner Sampler

---

**Require:** prior distribution $p_0$, number of integration steps $T$, and trained function $f_\theta$
  steps $\leftarrow 1$
  $\Delta t \leftarrow \frac{1}{T}$
  $t \leftarrow 0$
  $x_0 \sim p_0(x_0)$
  $x_t \leftarrow x_0$
  **while** steps $\leq T - 1$ **do**
    $\hat{x}_1 \leftarrow f_{\theta,t}(x_t)$
    $\hat{x}_1 \leftarrow f_{\theta,t}(\hat{x}_1)$
    $x_t \leftarrow x_t + \Delta t \cdot \frac{\hat{x}_1 - x_t}{1 - t}$
    $t \leftarrow t + \Delta t$
    steps $\leftarrow$ steps $+ 1$
  **end while**
  $x_1 \leftarrow x_t$

---

Interestingly, as the energy in Eq. 13 is lower bounded by 0, the convergence of the flow map is further guaranteed as:

$$f_\infty(x) \in \{x \in \mathbb{R}^n \mid \nabla_x E(x) = 0\} = \mathcal{M}, \quad (17)$$

where $\mathcal{M}$ is a subset of the equilibrium configurations. This implies that, starting from any $x \in \mathbb{R}^d$, the gradient flow will converge to some local minima of the energy function $E$, where the likelihood of the data is locally maximized.

### 3.3. Idempotent Flow Map

In general, the energy function defined in Eq. 13 renders $G$ to be a "neural refiner". To learn the refiner we can simply define the loss for $G$ as:

$$L_{\text{G}} = ||G(\hat{x}_1) - x_1||_2^2. \quad (18)$$

Intuitively, the refiner $G$ is easier to learn than the flow map $f$, and thus the flow map could have the capacity to *also refine* the mapped sample. This, interestingly, results in learning an *idempotent* flow map to its prediction:

$$f_{\theta,t}(x) = f_{\theta,t}(f_{\theta,t}(x)). \quad (19)$$

Hence, the energy landscape is shaped by mapping the off-the-manifold points $\hat{x}_1$ to on-manifold points $x_1$ (LeCun, 2022). The idempotence of the flow map is clear from the Eq. 17:

$$f_\infty(f_\infty(x)) = f_\infty(x), \quad (20)$$

i.e., iterating the flow map infinitely many times yields the same result. Under the assumption that our dataset of equilibrium configurations is contained in $\mathcal{M}$, we would always want to query $f_t(x)$ at $t = \infty$. Ideally, after training a model to learn the idempotent flow map, we would have $f_\infty(x) = x_1$ for any $x$. However, imperfections may remain, in which case $f_\infty(x) = \hat{x}$ would land somewhere close to $x_1$, but not exactly there. In this case, an iterative refinement procedure (Stärk et al., 2023; Jing et al., 2024), can also be applied during inference. The vector from $x$ to $\hat{x} = f(x)$ can be used as a step direction in an integration scheme, such as Euler's method. Heuristically, if the magnitude of the vector is large, the integrator will take a large step in that direction, and vice versa, eventually leading to stabilization around the final prediction.

Furthermore, enforcing the flow map $f_\theta$ to be idempotent draws an informative connection to the structure refinement regression model (Jumper et al., 2021) which recycles the output for iterative refinement. Hence, in tandem with the CFM loss, we propose our idempotent objective as:

$$L_{\text{R}}(\theta) := \mathbb{E}_{\substack{t \sim \mathcal{U}(0,1) \\ x \sim p_t(x|x_0,x_1) \\ \hat{x}_1 \sim \mathcal{N}(\hat{x}_1 | f_\theta(x,t), \sigma^2 I_d)}} \|f_\theta(\hat{x}_1.detach()) - x_1\|_2^2$$

$$(21)$$

where $\hat{x}_1$ is dynamically sampled from the flow model during training.

### 3.4. Training and Inference

The idempotent objective function enables the network to refine samples iteratively. Theoretically, the sample $\hat{x}_1$ could be refined for an infinite number of times. However, excessive refinements increase inference time, a key limitation of sampling-based methods. To mitigate this, we perform only one refinement per step. Specifically, the predictor-refiner sampler makes a prediction $\hat{x}_1$ and refines it, resulting in two

*Table 1.* SINGLE LIGAND DOCKING. Structure generation (ten samples average for each test example) comparison of methods on the PDBBind for pocket-level docking.

| METHOD | SEQUENCE SIMILARITY SPLIT | | | | TIME SPLIT | | | |
|---|---|---|---|---|---|---|---|---|
| | DISTANCE-POCKET | | RADIUS-POCKET | | DISTANCE-POCKET | | RADIUS-POCKET | |
| | %<2 | MED. | %<2 | MED. | %<2 | MED. | %<2 | MED. |
| PRODUCT SPACE DIFFUSION | 27.2 | 3.2 | 16.1 | 4.0 | 20.8 | 3.8 | 15.2 | 4.3 |
| HARMONICFLOW | 30.1 | 3.1 | 20.5 | **3.4** | 42.8 | 2.5 | 28.3 | 3.2 |
| **IDFLOW** | **35.6** | **2.9** | **21.0** | 3.7 | **44.3** | **2.4** | **34.7** | **3.1** |

*Table 2.* MULTI-LIGAND DOCKING. Structure generation (ten samples average for each test example) comparison of methods on the Binding MOAD.

| METHOD | $\% < 2$ | $\% < 5$ | MED. |
|---|---|---|---|
| EIGENFOLD DIFFUSION | 39.7 | 73.5 | 2.4 |
| HARMONICFLOW | **44.4** | 75.0 | 2.2 |
| **IDFLOW** | 43.8 | **83.1** | **2.1** |

**Number of Function Evaluations (NFEs) per step.** To better align training with sampling, we adopt a strategy similar to self-conditioning (Chen et al., 2023), where the training of the sampler and refiner is separated. For 50% of the training time, the network undergoes flow matching training. In the remaining 50%, the network first predicts $\hat{x}_1$ with the gradient detached, and then trains for the idempotent objective Eq. 21. Empirically, tuning the maximum number of iterations $K_{\max}$ helps achieve gradual refinement. Increasing $K_{\max}$ beyond 2 provides diminishing returns, with smaller variations observed at $K_{\max} = 2$. For the training of 'refiner', ideally, the *timestep* $t$ should be set to 1 as $\hat{x}_1$ represents data, but in practice we find it doesn't impact the method's performance. This approach also reduces memory overhead, as only $K - 1$ outputs need to be stored when looping the network $K$ times. The entire approach only introduces one extra hyperparameter $K_{max}$ and remains to be highly orthogonal to the existing methods. Further details are provided in Algorithms 1 and 2.

### 3.5. Architecture

For the docking task, the network is parameterized as the refinement tensor field network (Stärk et al., 2023; Thomas et al., 2018), predicting the ligand atom 3D positions $\mathbb{R}^{3 \times n_l}$. The building block for equivariance is the tensor product convolution layer, which constructs the message as the tensor product between the node and edge embedding. Besides, each layer is designed to predict the update of the previous layer through the auxiliary loss. For protein backbone generation, we leverage the invariant point attention (IPA)-based structure transformer (Jumper et al., 2021) to predict the protein frames. The IPA differs from the standard self-

attention layer with the attention weights to be invariant to the structure input. The architecture is kept the same as the baseline to maintain a fair comparison.

## 4. Experiments

In this section, we investigate several research questions about the proposed framework. **Q1**: How does *IDFlow* perform compared with the standard flow matching setup? **Q2**: Does the training setup also apply to Riemannian flow matching? **Q3**: Is *IDFlow* transferable for generating molecules of various sizes (both ligand and protein)? **Q4**: How does *IDFlow* perform for sampling the modes of distribution? **Q5**: How do different setups affect the *IDFlow*? To answer these questions, we design experiments for two widely applicable tasks, molecular docking (**Q1, Q4, Q5**) and protein backbone generation (**Q2, Q3, Q5**), which target predicting the binding pose of the ligand in a protein-ligand complex and generating physically plausible protein structures. Specifically, we build the proposed IDFlow from two recent flow matching-based structure generation models (Stärk et al., 2023; Yim et al., 2023a).

### 4.1. Molecular Docking

**Dataset and baselines.** To evaluate the structure generation capability of *IDFlow* for pocket-level docking, we train the model and its ablations on the PDBBind v2020 dataset (Liu et al., 2017) for both the time and the 30% sequence similarity split and the Binding MOAD (Hu et al., 2005) with 30% sequence similarity split, as proposed in (Stärk et al., 2023). We follow the dataset preprocessing steps of the HarmonicFlow for both PDBBind and BindingMOAD (details in E.1.1). For both single and multi-ligand docking, we consider HarmonicFlow (Stärk et al., 2023) and the product space diffusion model over ligand geometry (rotation, translation and torsion angles) (Corso et al., 2023) as the baselines. HarmonicFlow is the standard flow matching setup which samples the harmonic prior for ligand atoms and learns the flow map for predicting the bound structure of ligand. The reported results are averaged over three runs.

**Sampling time.** While the number of sampling steps sig-

*Table 3.* PROTEIN BACKBONE GENERATION RESULTS ON PDB. Comparison of methods for 200 generated proteins each at length [100, 150, 200, 250, 300]. The mean and standard deviation are reported over three runs for GAFL and FrameFlow. The results of FoldFlow2 are from the public weight, and other results are extracted from (Wagner et al., 2024). The number in the parentheses of FrameFlow and IDFlow is the NFEs for generating one sample. The time required to generate a backbone of length 100 is also reported. * Pretrained weights from folding model trained on a dataset larger than PDB.

| METHOD | DESIGNABILITY ($\uparrow$) | DIVERSITY ($\downarrow$) | NOVELTY ($\downarrow$) | HELIX CONTENT | STRAND CONTENT | TIME [S] |
|---|---|---|---|---|---|---|
| PDB DATASET (300) | - | - | - | 0.39 | 0.23 | - |
| FRAMEDIFF | 0.54 | 0.45 | 0.71 | 0.53 | 0.20 | 24.3 |
| FOLDFLOW-SFM | 0.69 | 0.44 | 0.77 | 0.91 | 0.01 | 24.3 |
| FOLDFLOW-OT | 0.82 | 0.44 | 0.79 | 0.88 | 0.00 | 24.3 |
| FOLDFLOW2 | **0.94** | 0.37 | **0.69** | 0.87 | 0.01 | **2.8** |
| RFDIFFUSION* | 0.89 | 0.37 | 0.74 | 0.58 | 0.24 | 21.0 |
| GAFL | $0.866 \pm 0.021$ | $0.35 \pm 0.01$ | $0.71 \pm 0.01$ | $\mathbf{0.53 \pm 0.01}$ | $\mathbf{0.24 \pm 0.01}$ | 8.8 |
| FRAMEFLOW (200) | $0.824 \pm 0.037$ | $0.35 \pm 0.01$ | $0.70 \pm 0.00$ | $0.57 \pm 0.01$ | $0.19 \pm 0.00$ | 6.6 |
| **IDFLOW** (100) | $0.871 \pm 0.013$ | $\mathbf{0.35 \pm 0.00}$ | $0.70 \pm 0.02$ | $0.60 \pm 0.07$ | $0.16 \pm 0.06$ | 3.3 |
| **IDFLOW** (200) | $0.927 \pm 0.020$ | $0.35 \pm 0.01$ | $0.72 \pm 0.01$ | $0.61 \pm 0.08$ | $0.16 \pm 0.07$ | 6.6 |

nificantly impacts performance, we maintain a consistent computational budget with HarmonicFlow, using 10 steps (20 NFEs), compared to HarmonicFlow's 20 steps.

**Evaluation metrics.** Following (Stärk et al., 2023; Corso et al., 2023), we use the fraction of the test samples that have root mean squared deviation (RMSD) below 2 or 5Å ($\% < 2$ and $\% < 5$) and the RMSD median (Med.) for evaluating the docking performance.

**Results.** We first investigate the method's performance on the single ligand docking and report metrics in Table 1. IDFlow consistently outperforms or is on par with both HarmonicFlow and product space diffusion models at the same inference time. Notably, IDFlow achieves 5.5% performance increase on RMSD $< 2$Å for sequence similarity split distance pocket docking and 6.4% on RMSD $< 2$Å for the time split radius pocket docking, demonstrating the improvement over different problem definition and datasets. Table 6 (App. F) shows great improvements in top-40, top-10, and top-5 accuracy for both RMSD $< 1$Å and RMSD $< 2$Å, highlighting the effectiveness of the approach for enhanced mode coverage. Table 2 shows the results on multi-ligand docking. Also in this setting, IDFlow maintains the same level of performance with HarmonicFlow on RMSD $< 2$Å, and demonstrates 8.1% performance increase for RMSD $< 5$Å.

### 4.2. Protein Backbone Generation

**Datasets and baselines.** In this section we evaluate *ID-Flow* on the task of unconditional protein structure generation. The experiments are first conducted on a small curated dataset SCOPe (Fox et al., 2014; Chandonia et al., 2022) comprised of 3928 protein structures filtered by lengths between 60 and 128 residues. Next, we evaluate the ID-Flow on the subset of PDB, with maximum protein length 512 and maximum coil content of 50 % filtering process

following (Yim et al., 2023b). This results in total 19327 proteins of various lengths for training. The main baseline is FrameFlow (Yim et al., 2023a) which we extend to build *IDFlow*. FrameFlow is the standard Riemnanian flow matching setup which samples from the source distribution to be the composition of uniform distribution on $SO(3)$ and standard Gaussian and learns the flow map to predict the protein structure. The results on SCOPe can be found in Table 7 (App. F). Again, to keep the sampling time the same as the baseline, we run inference with *IDFlow* using 50 steps (100 NFEs) on SCOPe and 100 steps (200 NFEs) on PDB.

**Evaluation metrics.** We adopt the commonly used metrics *designability, diversity* and *novelty* as in (Wu et al., 2024; Yim et al., 2023b; Bose et al., 2024; Huguet et al., 2024; Wagner et al., 2024). Designablility can be seen as a proxy for biophysical consistency and reports whether an amino acid sequence can be found that folds into the generated protein structure. Diversity and novelty report how structurally different the generated structures are to each other, and to the existing protein structures, respectively. For more detailed definition of metrics we refer to App. E.2.1.

**Results on PDB.** Table 3 summarizes the experiments' results with models trained on the PDB dataset. With 100 NFEs (50 steps), IDFlow achieves 5.5% performance increase in designability over FrameFlow while being competitive at diversity and novelty. With 200 NFEs (100 steps), IDFlow obtains designability with 10.3% increase and even approximates the performance of FoldFlow2, which relies on a more advanced architecture, data curation, and is conditioned on sequence information, on most metrics. Remarkably, in contrast to FoldFlow2, IDFlow does not tend to collapse to generating only helical protein structures. Fig. 2 visualizes a set of randomly selected designable protein structures generated by IDFlow. Proteins are arranged in non-redundant topologies and are comprised by both $\alpha$-

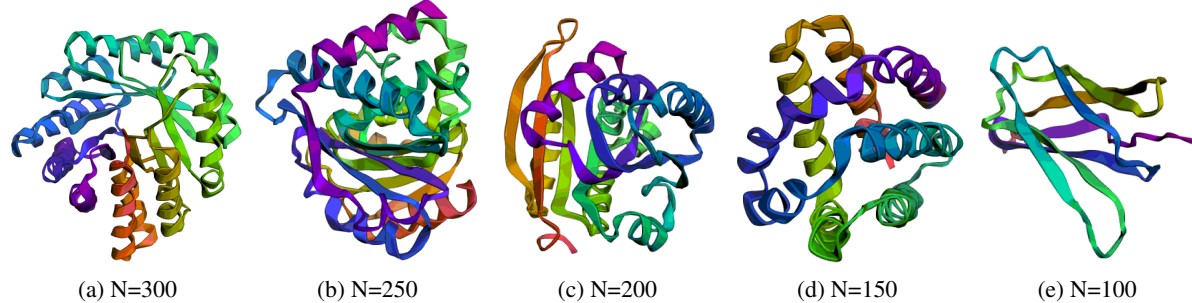

| (a) N=300 | (b) N=250 | (c) N=200 | (d) N=150 | (e) N=100 |

*Figure 2.* Designable protein backbones generated by IDFlow at various length N = {100, 150, 200, 250, 300}.

*Table 4.* Ablation on timesplit radius pocket docking.

|  | $\% < 2$ | MED. |
|---|---|---|
| $K_{max} = 0$ (HARMONICFLOW) | 28.3 | 3.2 |
| $K_{max} = 1$ | $34.2 \pm 2.42$ | $3.0 \pm 0.05$ |
| $K_{max} = 2$ | $34.7 \pm 0.17$ | $3.1 \pm 0.09$ |

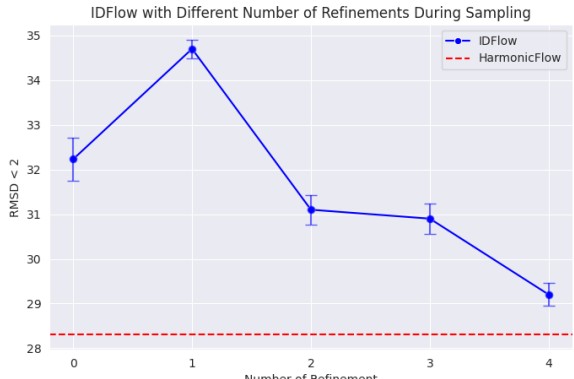

*Figure 3.* Ablation on the number of refinements $k$ at test time inference.

helices and $\beta$-strands.

### 4.3. Ablations

**Number of iterations.** We first ablate the number of iterations $K_{max}$ training for idempotency on the timesplit radius pocket docking in Table 4 for IDFlow. A single iteration improves over a baseline, with more iterations resulting in marginal performance gains but less variation. Fig. 3 shows the ablation of number of refinements of sampling at test time. Since more refinements increase the total number of NFEs, we keep the overall NFEs around 20. The performance peaks at one refinement. We attribute the performance drop with more refinement steps to the discretization errors of the ODE.

## 5. Related Work

**Generative modeling for molecular structure generation.** Generative modeling, mostly diffusion and flow matching models have been applied to generate molecular structures of ligands (Stärk et al., 2023) and proteins (Yim et al., 2023b;a). Molecular conformer generation (Xu et al., 2022a; Jing et al., 2022; Wang et al., 2024; Hassan et al., 2024) maps the molecular graph to the 3D position of atoms by learning the posing distribution. Sampling-based docking method (Corso et al., 2023; 2024; Qiao et al., 2024; Lu et al., 2024) models the bound structure of ligand given the receptor structure. Several generative models for protein structure (Watson et al., 2023; Wu et al., 2024; Campbell et al., 2024; Bose et al., 2024; Huguet et al., 2024; Yim et al., 2023b;a; Wagner et al., 2024) have been shown to sample physically plausible and functional protein structures *de novo*.

## 6. Conclusion

We present an energy-based formulation for flow matching, an enhanced training framework for flow matching combined with the EBMs for 3D molecular structure generation. We provide a specific instance of the proposed framework, IDFlow, which considers the reconstruction error as the energy function, shaping the loss landscape with contrastive samples produced by the flow model. It builds the flow map to predict and refine the sampling trajectory iteratively toward the target structure. We validate the improvements of IDFlow over existing flow matching methods in both Riemannian and Euclidean spaces as an alternative to the standard flow matching training and sampling paradigm, demonstrating its effectiveness over task-associated flow matching and diffusion model baselines in molecular docking and protein backbone generation.

One limitation of the work is the increased training cost, as multiple forward passes are required. Besides, extra refinement incurs a larger discretization error for the ODE sampling. Future work can combine the EBMs with biophysics-informed energy to generate chemically plausible structures and use EBMs as scoring functions for realistic docking.

## Impact Statement

This paper presents work whose goal is to advance the field of Machine Learning. There are many potential societal consequences of our work, none which we feel must be specifically highlighted here.

## Acknowledgements

This work was partially supported by the Knut and Alice Wallenberg Foundation. The computations were enabled by resources provided by the National Academic Infrastructure for Supercomputing in Sweden (NAISS), partially funded by the Swedish Research Council through grant agreement no. 2022-06725. Experiments were performed using the supercomputing resource Berzelius provided by the National Supercomputer Centre at Linköping University and the Knut and Alice Wallenberg foundation.

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

## Appendix Overview

In the Appendix, we provide the details:

- In section B, More discussions including the idempotency and stability, additional energy function, the general format of vector field with data parameterization flow matching and the associated energy of reconstruction error are provided.

- In section G, we visualize the generated samples.

## A. More Related Work

**Flow matching.** Flow matching (Lipman et al., 2023; 2024; Albergo & Vanden-Eijnden, 2023; Liu et al., 2022) is a simulation-free training paradigm of continuous normalizing flows (Chen et al., 2018) generalizing the diffusion models into arbitrary priors and flexible construction of probability paths. It builds the transformation from source to target distribution via an ODE and supports the source target sample coupling through optimal transport within batch samples (Tong et al., 2024; Pooladian et al., 2023). This approach can also be extended to manifold where the trajectory simulation is avoided on simple geometry (Chen & Lipman, 2024).

**Sampling-based molecular docking.** The seminal work DIFFDOCK (Corso et al., 2023) defines the diffusion model on the ligand geometry, specifically, the product space of global rotation, translation and torsion angles of ligand for efficiently modeling the docking conformation of ligand given the protein. (Corso et al., 2024) further investigates the method's scalability and uses confidence bootstrapping for enhanced training. DYNAMICBIND (Lu et al., 2024) extends the existing approach to the flexibility of protein backbone and side chains, modeling the distribution shift from apo to holo structure. NEURALPLEXER (Qiao et al., 2024) is a co-folding approach combining the diffusion models and structure prediction model for sampling the protein-ligand complex. Recently, to align with the realistic docking scenario, the diffusion-based docking is constrained to be within the pocket with the side chain angles flexibility (Plainer et al., 2023; Huang et al., 2024). HARMONICFLOW (Stärk et al., 2023) is the first flow matching method applied to the pocket-level docking for generating 3D structure by directly modeling the Cartesian coordinates of ligand. FLEXDOCK (Corso et al., 2025) proposes an unbalanced flow matching framework to ease the learning of large conformational changes of protein and chain the flow for generating the energetically valid poses.

**Protein backbone generation.** Generating protein backbone has risen to be promising with recent diffusion (Wu et al., 2024; Yim et al., 2023b; Watson et al., 2023) and flow matching-based methods (Yim et al., 2023a; Bose et al., 2024; Huguet et al., 2024; Wagner et al., 2024). RFDIFFUSION (Watson et al., 2023) achieved significant success for biologically validated structures and functions in experiments. FOLDFLOW (Bose et al., 2024) and FRAMEFLOW (Yim et al., 2023a) adopted the protein frame representation for generation with $SE(3)$ flow matching. FOLDFLOW2 (Huguet et al., 2024) extends the method with sequence input, architecture improvement and the Reinforced Finetuning on secondary structure. GAFL (Wagner et al., 2024) improves upon the FRAMEFLOW with Clifford frame attention rather than the IPA.

## B. More Discussions

### B.1. Idempotency and Stability

Idempotency and stability are two fundamental notions over many disciplines, for example, formation control, dynamical systems and molecular dynamics. It has been recently explored for sampling from stable distribution (Sprague et al., 2024) or being used as the building block for generative models (Shocher et al., 2024). It potentially has a broader impact in many scientific disciplines, not limited to generating molecules, but also structure elucidation (Cheng et al., 2024), etc.

Idempotency helps learn a robust function where additional function evaluations do not drastically alter the output. This relates to stability in terms of the loss used during training such that the iteratively generated output stays on the data manifold, a desirable property for any generative model. Moreover, the concept of stability could be further expanded into any energy function. Combined with flow models, existing work (Xu et al., 2022b; 2023) proposed to learn a stable vector field to be a Poisson field implicitly governed by the potential energy induced by the training sample over the augmented space of the data. Consistency models (Song et al., 2023) achieves fast sampling with just one or two steps of denoising by learning a consistent function across multiple noise levels. The idempotency is only guaranteed at $t = 0$ by design as the boundary conditions.

### B.2. Other Energy Function

IDFlow proposes to use the same energy function as the CFM loss in the context of flow matching which resulting in a relatively smooth minimum in the loss landscape. In general, any type of energy function can be considered.

**Distance potential.** The relative distances among atoms deliver important information for structure prediction (Senior et al., 2020). Assuming a certain distribution on the distance can help derive a statistical approximation of the energy function (Méndez-Lucio et al., 2021). Specifically, the intramolecular potential enforces constraints on the ligand atoms pairs to be within certain ranges of distances, depending on the bond type connecting the atoms. For the protein-ligand complex, the intermolecular potential sets up the constraint of the distance between the ligand atoms and protein residues within the binding pocket. Hence, we can enforce the preservation of the protein-ligand distance and the bond distance of the ligand as an energy function:

$$E(\hat{x}_1) = \sum_{(i,j)\in\text{bond}} ||d_{i,j} - \hat{d}_{i,j}|| + \sum_{(i,j)\in\text{pocket protein ligand atom pairs}} ||d_{i,j} - \hat{d}_{i,j}|| \tag{22}$$

**Machine learning force fields.** MLFF (e.g, openmm or fair-chem) can be applied to sample relaxation so that the optimized samples is energetically favorable. However, sample relaxation may lead to larger RMSD error as the projected gradient $\nabla_{x_1} E(x_1)$ ($x_1$ refers to the data) may leave the data manifold (Ben-Hamu et al., 2024). Within the proposed framework, we can apply the MLFF at each denoising step:

$$\hat{x}_1 = f_{\theta,t}(x_t)$$
$$\hat{x}_1 = \text{FF}(\hat{x}_1) \quad \text{or} \quad \hat{x}_1 = \hat{x}_1 - \frac{\epsilon^2}{2}\nabla_{\hat{x}_1} E(\hat{x}_1) + \epsilon z$$
$$x_{t+\Delta t} = x_t + \frac{\hat{x}_1 - x_t}{1 - t}$$

Throughout the iterative procedure of sample prediction and relaxation at each step, intuitively, a low energy valid sample can be generated. However, this is prohibitive in practice due to the huge increase in sampling time of running the MLFF, and thus one may only use the MLFF when the sample path is close to the data or use a neural refiner to approximate the low energy sample, such as the structure relaxation flow trained with the flat bottom potential (Corso et al., 2025). Another option is to sample from the EBM through the Langevin MCMC, which intends to sample from the true distribution governed by the energy function $\frac{1}{Z_\theta}(\exp(-E_\theta(\hat{x}_1)))$. The $\epsilon$ is the step size and $z$ is the standard Gaussian sample.

### B.3. Vector Field of the Data Parameterization

With the data parameterization of flow matching, the exact format of the vector field $v_t(x)$ could be diverse as long as the predicted data $\hat{x}_1 = f_{\theta,t}(x)$ could be reached following this vector field. As the trajectory sample $x_t$ is often constructed as a linear interpolation, the conditional vector field that generates a single sample at time $t$ is linear in the following general format:

$$u_t(x|x_1, x_0) = A(t)(x - x_1) \tag{23}$$

where $A(t)$ is a time-dependent rescaling coefficient. As the marginal vector field $u_t(x)$ is the convex combination of the conditional vector fields (Eq. 8 from (Lipman et al., 2023)), we can directly plug into the computation:

$$
\begin{aligned}
u_t(x) &= \iint u_t(x|x_1, x_0) \frac{p_t(x|x_1, x_0)q(x_1)p(x_0)}{p_t(x)} \mathrm{d}x_1 \mathrm{d}x_0 \\
&= \iint A(t)(x - x_1) \frac{p_t(x|x_1, x_0)q(x_1)p(x_0)}{p_t(x)} \mathrm{d}x_1 \mathrm{d}x_0 \\
&= A(t) \iint (x - x_1) p(x_1, x_0|x) \mathrm{d}x_1 \mathrm{d}x_0 \\
&= A(t) \int (x - x_1) p_{1|t}(x_1|x) \mathrm{d}x_1 \\
&= A(t) \mathbb{E}_{x_1 \sim p_{1|t}(x_1|x)} [x - x_1 | x_t = x] \\
&= A(t) \left( x - \mathbb{E}_{x_1 \sim p_{1|t}(x_1|x)} [x_1 | x_t = x] \right)
\end{aligned}
$$

where in the third equality we use the Bayes rule assuming $q(x_1)$ and $p(x_0)$ are independent. This yields a general format of the marginal vector field supposing the conditional vector field is linear. In practice, the flow map $f_{\theta,t}(x)$ is trained to predict the data, implicitly learning the expected value of $x_1$ given $x$. Hence, the form of the marginal vector field with data parameterization is:

$$
\begin{aligned}
u_t(x) &= A(t)(x - \mathbb{E}_{x_1 \sim p_{1|t}(x_1|x)} [x_1 | x_t = x] \\
&\approx A(t)(x - f_{\theta,t}(x))
\end{aligned}
$$

With $A(t) = -\frac{1}{1-t}$, the marginal vector field trained by the conditional optimal transport is recovered as:

$$
u_t(x) = \frac{f_{\theta,t}(x) - x}{1-t}. \tag{24}
$$

Interestingly, with the learned flow map as the expectation over $p_{1|t}(x_1|x)$, the flow map's idempotency is clear as the expectation is an idempotent operator:

$$
f_{\theta,t}(f_{\theta,t}(x)) = \mathbb{E}[\mathbb{E}_{x_1 \sim p_{1|t}(x_1|x)} [x_1 | x_t = x]] = \mathbb{E}_{x_1 \sim p_{1|t}(x_1|x)} [x_1 | x_t = x]. \tag{25}
$$

### B.4. Conditional Negative likelihood Energy

Conditional flow matching assumes a smooth delta function for the training example $x \sim \mathcal{N}(x|x_1, \sigma_1^2 I_d)$:

$$
p(x|x_1) \sim \exp \left( -\frac{1}{2\sigma_1^2} (x - x_1)^\top (x - x_1) \right) \tag{26}
$$

where $x_1$ is a *single* sample from the training set. However, the true $x_1$ is unknown during sampling, and $\hat{x}_1 = f_{\theta,t}(x)$ is estimated to refine the trajectory of the sampling path. We can approximately assume the $\hat{x}_1$ distribution follows a similar format:

$$
p(\hat{x}_1|x_1) \sim \exp \left( -\frac{1}{2\sigma_1^2} (\hat{x}_1 - x_1)^\top (\hat{x}_1 - x_1) \right) \tag{27}
$$

Ideally, each sampling step intends to find an estimate of $\hat{x}_1$ such that it lies in the high-density region and approximates to be stable around a certain area of the energy function. With the exact energy format as the negative log-likelihood of Eq. 27, the energy of $\hat{x}_1$ could be computed as:

$$
E(\hat{x}_1) = -\log p(\hat{x}_1|x_1) = \frac{1}{2\sigma_1^2} (\hat{x}_1 - x_1)^\top (\hat{x}_1 - x_1) \tag{28}
$$

Furthermore, considering the energy minimum $x'$ of Eq. 28 is equivalent to the gradient of the energy function equal to zero, and we could obtain the $x'$ by a neural network $G(\hat{x}_1)$ which could be seen as a neural refiner that projects the $\hat{x}_1$ to its associate point on the data manifold. The gradient of the likelihood of $\hat{x}_1$ could be written as:

$$
G(\hat{x}_1) = x' \qquad \nabla E(\hat{x}_1) = \nabla \left( \frac{1}{2\sigma_1^2} (x' - x_1)^\top (x' - x_1) \right) = -\nabla \log p(\hat{x}_1|x_1) = \frac{1}{\sigma_1^2} (\nabla G(\hat{x}_1))(G(\hat{x}_1) - x_1) = 0 \tag{29}
$$

Observing Eq. 29, the term could be optimized if the neural refiner $G$ approximates the $x_1$, which aligns with the proposed idempotent objective 21:

$$L_{\text{R}} = ||G(\hat{x}_1) - x_1||^2. \tag{30}$$

## C. An Overview of Riemannian Flow Matching

### C.1. Riemannian Manifold

Considering a smooth manifold $\mathcal{M}$ that is locally Euclidean (i.e., each point has a neighborhood that can be smoothly mapped to an open subset of $\mathbb{R}^n$) and differentiable, each point of the manifold $x \in \mathcal{M}$ has a neighborhood that resembles an open subset of $\mathbb{R}^n$. The tangent plane attached to each point $x$ on the manifold is called the *tangent space* $\mathcal{T}_x\mathcal{M}$. The collection of all the tangent spaces $\mathcal{T}_x\mathcal{M}$ for each point $x \in \mathcal{M}$ forms the *tangent bundle* of the manifold, denoted by $T\mathcal{M}$.

The Riemannian metric $g$ is a smoothly varying inner product on the tangent spaces of the manifold. Formally, at each point $x \in \mathcal{M}$, the tangent space $\mathcal{T}_x\mathcal{M}$ is equipped with a positive-definite inner product $\langle \cdot, \cdot \rangle_x$ varying across the manifold. This is typically denoted as $g_x$ for measuring the distance on the manifold. For instance, the metric on $\text{SO}(3)$ typically requires a bilinear function $\langle \cdot, \cdot \rangle : \mathbb{R}^3 \times \mathbb{R}^3 \to \mathbb{R}$ to be symmetric positive definite. Hence, we could define a positive definite quadratic form for the metric:

$$\langle \mathfrak{r}_1, \mathfrak{r}_2 \rangle_{\text{SO}(3)} = \frac{1}{2}\text{tr}(\mathfrak{r}_1^\top \mathfrak{r}_2) \tag{31}$$

where $\mathfrak{r}_1$ and $\mathfrak{r}_2$ are elements of the Lie algebra of $\text{SO}(3)$, which is the tangent space at the identity element of $\text{SO}(3)$ and consists of skew-symmetric matrices.

*Geodesic* generalizes the concept of a straight line in Euclidean space to a Riemannian manifold, where it locally minimizes the distance between two points on the manifold. It preserves the Riemannian metric and allows for parallel transport of vectors along curves.

The *exponential map* $\exp_x(v) : \mathcal{T}_x\mathcal{M} \to \mathcal{M}$ takes a tangent vector $v$ on $\mathcal{T}_x\mathcal{M}$ and maps it to another point $y$ on the manifold along the geodesic. The inverse of the exponential map is called the *logarithm map*, denoted as $\log_x(y) : \mathcal{M} \to \mathcal{T}_x\mathcal{M}$. The logarithm map takes a point $y \in \mathcal{M}$ and returns the tangent vector at $x$ that connects $x$ and $y$ through the geodesic distance.

### C.2. Riemannian Flow Matching on Protein Geometry

As stated in 3.1, the Riemannian flow matching on protein geometry is defined on the $\text{SE}(3)^N$ group. This backbone parameterization, proposed by AlphaFold2 (Jumper et al., 2021), denotes a frame representation $\text{SE}(3)$ for each protein residue. With $N$ amino acids, the protein backbone lies on the manifold $\text{SE}(3)^N$. Each frame $x = (r, s) \equiv \text{SE}(3)$ consisting of a rotation $r$ and translation $s$ applies the rigid transformation to the reference frame. The reference frame has four backbone atoms with idealized coordinates centered at the $C_\alpha^*$ atom:

$$N^\star = (-0.525, 1.363, 0.0)$$
$$C_\alpha^\star = (0.0, 0.0, 0.0)$$
$$C^\star = (1.526, 0.0, 0.0)$$
$$O^\star = (0.627, 1.062, 0.0)$$

The idealized coordinates are the experimental measurement of bond angles and lengths (Engh & Huber, 2012).

$\text{SE}(3)^N$ Riemannian flow matching requires to devise a metric on $\text{SE}(3)$. As the $\text{SE}(3)$ is a semi-direct group, one suitable metric can decompose the computation into $\text{SO}(3)$ and $\mathbb{R}^3$: $\langle \mathfrak{r}_1, \mathfrak{r}_2 \rangle_{\text{SE}(3)} = \langle \mathfrak{r}_1, \mathfrak{r}_2 \rangle_{\text{SO}(3)} + \langle \mathfrak{r}_1, \mathfrak{r}_2 \rangle_{\mathbb{R}^3}$. This results in the geodesic of $\text{SE}(3)$ becomes the same as those on the product space of $\text{SO}(3)$ and $\mathbb{R}^3$. Moreover, as the Lie algebra of $\text{SO}(3)$ is a skew-symmetric matrix, its matrix exponential has a closed-form solution, referred to as the Rodrigues formula. Therefore, its exponential and logarithms map could be computed efficiently without simulating the trajectory over the manifold. In particular, assuming $\mathbf{w}$ is a rotation vector, known as the axis-angle representation, and $\hat{w}$ is its uniquely identified element of the Lie algebra, its matrix exponential and logarithms can be expressed as (Bose et al., 2024):

$$\textbf{Matrix Exponential:} \quad R = \exp(\hat{w}) = \cos(w)I + \sin(w)e_w + (1 - \cos(w))e_w e_w^\top \tag{32}$$

$$\textbf{Matrix Logarithms:} \quad \hat{w} = \log(R) = \frac{w}{2\sin(w)}(R - R^\top) \tag{33}$$

where $w = ||\mathbf{w}||$, $e_w = \frac{\mathbf{w}}{||\mathbf{w}||}$ represents the angle and axis of the rotation, and $R$ is the SO(3) group element on the manifold. From Eq. 33, the angle $w$ could be derived from the trace of the rotation matrix $R$: $\cos(w) = \frac{\text{tr}(R)-1}{2}$. For the translation $\mathbb{R}^3$, the computation is trivial following the Euclidean formulation.

With the matrix exponential and logarithm on hand, we can construct the linear interpolant on SE(3) as the Riemannian conditional optimal transport:

$$\textbf{Translation:} \quad s_t = ts_1 + (1-t)s_0 \tag{34}$$

$$\textbf{Rotation:} \quad r_t = \exp_{r_0}(t\log_{r_0}(r_1)) \tag{35}$$

where $x = (r, s)$ is the frame. Then, the flow matching objective extended to the $\text{SE}(3)^N$ is:

$$L_{\text{SE}(3)}(\theta) := \mathbb{E}_{\substack{t \sim \mathcal{U}(0,1) \\ x \sim p_t(x|x_0,x_1) \\ x_0 \sim p(x_0) \\ x_1 \sim q(x_1)}} \left[ \sum_{n=1}^{N} \left( \|v_{\theta,t}(s) - \dot{s}_t\|_{\mathbb{R}^3}^2 + \|v_{\theta,t}(r) - \dot{r}_t\|_{\text{SO}(3)}^2 \right) \right] \tag{36}$$

where $x_0$ samples from the uniform distribution on SO(3) and Gaussian on $\mathbb{R}^3$, and $x_1$ samples from the training set.

$x_1$ **parameterization.** The objective can also be reparametrized as the flow map directly predicting $x_1 = (s_1, r_1)$:

$$L_{\text{SE}(3)}(\theta) := \mathbb{E}_{\substack{t \sim \mathcal{U}(0,1) \\ x \sim p_t(x|x_0,x_1) \\ x_0 \sim p(x_0) \\ x_1 \sim q(x_1)}} \left[ \sum_{n=1}^{N} \left( \|f_{\theta,t}(s) - s_1\|_{\mathbb{R}^3}^2 + \|f_{\theta,t}(r) - r_1\|_{\text{SO}(3)}^2 \right) \right] \tag{37}$$

## D. Architecture Details

The architecture used for the IDflow is kept to be the same as the main baseline HARMONICFLOW and FRAMEFLOW, including node and edge features initialization. We summarize the architecture details here.

### D.1. Refinement Tensor Field Networks

Tensor Field Networks (TFN) (Thomas et al., 2018) incorporates the equivariance by parameterizing the node feature update as the path weight of the tensor product $\otimes_w$ between scalar node features $h_i$ and spherical harmonics of the edge vector. Denoting the invariant feature of the node $i$ as $h_i$ and coordinate as $x_i$, the edge vector connecting the node $i$ and $j$ is $\hat{r}_{ij} = x_i - x_j$. The node update is built as:

$$\psi_{ij} = \Psi(e_{ij}, h_i^k, h_j^k)$$
$$m_{ij} = Y(\hat{r}_{ij}) \otimes_{\psi_{ij}} h_j^k$$
$$h_i^{k+1} = h_i^k + \text{BN}\left( \frac{1}{|N_i|} \sum_{j \in N_i} m_{ij} \right)$$

We first compute the path weights $\psi_{ij}$ as the coefficients of the spherical harmonics, where $e_{ij}$ is the edge embedding between node $i$ and node $j$, the superscript $k$ in $h_i^k$ represents node feature of the $k^{th}$ layer, and $\Psi$ is the linear layer defined separately for four different types of edges (receptor to receptor, ligand to ligand, ligand to receptor, receptor to ligand). Then the message $m_{ij}$ is constructed by the tensor product, in which $Y(\hat{r}_{ij})$ is the evaluation of the spherical harmonics at $\hat{r}_{ij}$. All the aggregated messages are averaged over the number of neighbours and then pass to the equivariant batch normalization.

Each layer first updates the node feature, and then updates the coordinate via the $O(3)$ equivariant linear layer $\Phi$:

$$x^{k+1} \leftarrow \hat{x}^{k+1} + \Phi(h^{k+1}) \tag{38}$$

The learning paradigm involves feature and coordinate refinement, where each layer of the network predicts updates for the previous layer. Additionally, the coordinate output at each layer is "deeply supervised" by the training annotations through the auxiliary loss function. The implementation is based on the e3nn library (Geiger & Smidt, 2022).

## D.2. IPA-Based Structure Transformer

The invariant point attention mechanism for equivarance and backbone parameterization are initially proposed by (Jumper et al., 2021). The entire procedure involves the network prediction and the backbone update. More details can be found in (Yim et al., 2023b) of Appendix I. We list the algorithms here:

**Node update.** Each layer consists of an IPA embedding layer, a transformer layer and a linear layer for feature transition.

$$
\begin{aligned}
h_{\text{ipa}} &= \text{IPA}(h_\ell, z_\ell, T_\ell) \\
h_{\text{in}} &= \text{LayerNorm}(h_{\text{ipa}} + h_\ell) \\
h_{\text{trans}} &= \text{Transformer}(h_{\text{in}}) \\
h_{\text{out}} &= \text{Linear}(h_{\text{trans}}) + h_\ell \\
h_{\ell+1} &= \text{MLP}(h_{\text{out}})
\end{aligned}
$$

**Edge update.** The edge update is computed by first projecting the updated node feature to half the dimension $h_{\text{down}}$. The projected features is then concatenated with the edge embedding feeding into the linear layer:

$$
\begin{aligned}
h_{\text{down}} &= \text{Linear}(h_{\ell+1}) \\
z_{\text{in}}^{ij} &= \text{cat}(h_{\text{down}}^n, h_{\text{down}}^m, e_\ell^{ij}) \\
z_{\ell+1} &= \text{LayerNorm}(\text{Linear}(z_{\text{in}}))
\end{aligned}
$$

where $h_{\text{down}}^n$ and $h_{\text{down}}^m$ are features expanded into two separate dimensions.

**Backbone update.** The network output is structured to be translation update $s_{\text{update}}$ and the quaternion representation $(1, b_i, c_i, d_i)$ then being converted into a rotation matrix.

$$
\begin{aligned}
b_i, c_i, d_i, s_i^{\text{update}} &= \text{Linear}(h_\ell) \\
a_i, b_i, c_i, d_i &= (1, b_i, c_i, d_i)/\sqrt{1 + b_i^2 + c_i^2 + d_i^2} \\
R_i^{\text{update}} &= \begin{pmatrix} (a_i)^2 + (b_i)^2 - (c_i)^2 - (d_i)^2 & 2b_ic_i - 2a_id_i & 2b_id_i + 2a_ic_i \\ 2b_ic_i + 2a_id_i & (a_i)^2 - (b_i)^2 + (c_i)^2 - (d_i)^2 & 2c_id_i - 2a_ib_i \\ 2b_id_i - 2a_ic_i & 2b_id_i - 2a_ic_i & (a_i)^2 - (b_i)^2 - (c_i)^2 + (d_i)^2 \end{pmatrix} \\
x_i^{\text{update}} &= (R_i^{\text{update}}, s_i^{\text{update}}) \\
x_{\ell+1} &= x_\ell \cdot x_i^{\text{update}}
\end{aligned}
$$

where i is the index of protein residue.

# E. Experimental Details

In this section, we provide additional details about the datasets, experimental setup and evaluation metrics. The source code is available at https://github.com/CaviarLover/IDFlow.

## E.1. Molecular Docking

### E.1.1. DATASETS

The PDBBind v2020(Liu et al., 2017) with a total of 19k complexes timesplit is a commonly used benchmark for molecular docking (Stärk et al., 2022; Lu et al., 2022; Corso et al., 2023; Zhang et al., 2023; Pei et al., 2024; Corso et al., 2024). The time split proposed by (Stärk et al., 2022) consists of 17k complexes before 2019 for training and validation and 363 complexes after 2019 for testing without the seen ligand in the training set. The 30% sequence similarity split is constructed from the same dataset but with the constraint of the chain-wise similarity less than 30%, which is considered as a more difficult split for the timesplit.

BindingMOAD (Hu et al., 2005) is another curated dataset from the PDB, with a different preprocessing pipeline from the PDBBind, ending up with 41k complexes. The dataset has been recently explored for more challenging benchmark

construction and multi-ligand docking. Similar to the PDBBind, the maximum $30\%$ sequence similarity split provides 56649, 1136 and 1288 for training, validation and testing. Only one biounit for each complex is used for training. The complex with only one contact (protein residue ligand atom distance less than 4) is further filtered out, retrieving 36203, 734 and 756 training, validation and test examples.

### E.1.2. POCKET DEFINITION

**Radius Pocket.** The pocket center is the mean position of the protein residues of which the minimum distance to any ligand atoms is less than 8Å. The radius is computed as the maximum between 5Å and the ligand radius (half of the largest distance between the ligand atoms) plus the radius pocket buffer (set to 7Å in all experiments). The pocket residues are selected based on the comparison between the residue pocket center distance and the radius. The residue pocket center distance is randomly flipped by the $\sigma = 2$.

**Distance Pocket.** The protein residue ligand atoms distances are extracted by the minimum distance between the residue and any ligand atom positions. Again, these protein-ligand distances are further randomly shifted with the $\sigma = 2$. The final pocket residues are the ones whose distances are below 14 Å.

### E.1.3. EXPERIMENTAL SETUP

The number of vector features and scalar features for TFN is set to be 32 and 8, respectively. Hence, there is no higher order representation ($> 1$) being used in the experiment and we do not use batch normalization and residual connection for the aggregated messages, but only layernorm the input features for each layer. The batch is set to be 4 for each GPU. The model is trained for 150 epochs using the Adam optimizer (Kingma & Ba, 2014) with the initial learning rate 1e-3 and a polynomial scheduler. The flow matching conditional standard deviation is set to be constant $\sigma_t = \sigma = 0.5$. The training takes around 20 hours on 8 RTX A100 GPUs for single ligand docking and one day on multi-ligand docking. The validation is conducted for every epoch, and the checkpoint with the largest RMSD $<2$Åis selected for inference. The number of function evaluations is set to be 20 consistent with the HarmonicFlow. More details can be found in the GitHub repository of HarmonicFlow (https://github.com/HannesStark/FlowSite).

## E.2. Protein Backbone Generation

### E.2.1. EVALUATION METRICS

**Designability.** In our experiments we follow an established self-consistency evaluation pipeline introduced before (Trippe et al., 2022; Watson et al., 2023). For every sampled backbone, we perform inverse folding with ProteinMPNN (Dauparas et al., 2022) and generate 8 sequences, which are subsequently refolded with ESMfold (Lin et al., 2023). We then compute the self-consistency RMSD (scRMSD) by aligning each ESMfold-refolded candidate with the originally sampled backbone on $C_\alpha$ atoms and consider a backbone designable if scRSMD $\leq 2.0$ Å.

**Diversity.** This metric measures the similarity among the **designable** backbones by computing the pairwise Template Modeling (TM) score.

$$Diversity = \frac{1}{N} \sum_{l=1}^{N} \frac{1}{n_l(n_l - 1)} \sum_{i \neq j} \text{TM}(x_i, x_j)$$

where N is the total number of protein lengths, $n_l$ is the number of designable backbones at each length l, i and j are the index of the protein at each length.

**Novelty.** The novelty is defined as the TM score between the **designable** backbone and its closest natural protein found in the PDB databases with FoldSeek (Van Kempen et al., 2024), averaging over all the designable proteins:

$$Novelty = \frac{1}{n} \sum_{i=1}^{n} \max_j \text{TM}(x_i, x_j)$$

where n is the number of designable proteins.

### E.2.2. EXPERIMENTAL SETUP

We train the model on 8 RTX A100 GPUs for 150 epochs ($\sim$ 22 hours) on SCOPe and 600 epochs ($\sim$ 3 days) on PDB. After $N_{\text{train}}$ epochs of training, the checkpoints are swept every $N_{\text{sweep}}$ for inference. The checkpoint achieving the highest designability is selected for the final result. Respectively, $N_{\text{train}}$ is set to 100 and 300 for SCOPe and PDB, and $N_{\text{sweep}}$ is set to 10 and 50. The number of iteration $K_{max}$ is set to be 1, as any value beyond 1 fills out of the memory even on "fat" GPU. The other hyperparameters are kept to be the same as the FrameFlow (https://github.com/microsoft/protein-frame-flow). We report the key setting here:

| Hyperparameter | Setup |
|---|---|
| learning rate | 1e-4 |
| node embedding dimension | 256 |
| edge embedding dimension | 128 |
| number of head for IPA | 8 |
| number of query / key points | 8 |
| number of value channel | 12 |
| number of head for transformer layer | 4 |
| number of layer | 6 |

*Table 5.* Key hyperparameter setup for IDFlow on protein backbone generation.

## F. Additional Results

**Validation metrics on docking.**   Fig. 4 shows the validation metric curve of HarmonicFlow and IDFlow. IDFlow converges much faster and achieves higher RMSD $< 2$Å than HarmonicFlow.

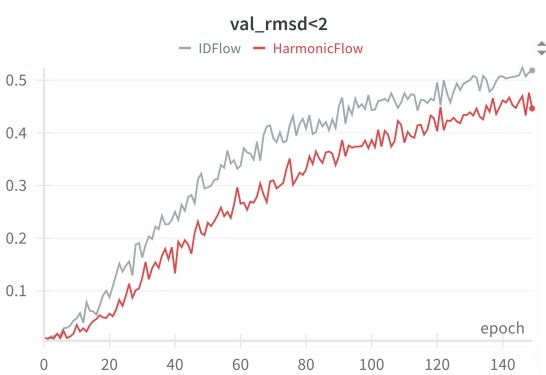

*Figure 4.* The validation metric curve of docking on PDBBind of radius pocket time split. The fraction of validation RMSD $< 2$Å vs. epoch time for IDFlow and HarmonicFlow.

**Top-k accuracy on docking.**   The top-**k** accuracy assumes the known ground truth and picks up the sample with the highest accuracy of "RMSD $< 2$". **k** is the number of samples being generated. It reflects the model's capability of mode sampling. Table 6 shows the results of *single ligand docking* on PDBBind. IDFlow consistently outperforms the HarmonicFlow on different metrics across the dataset, demonstrating the improved mode capturing of IDFlow.

*Table 6.* Top-k (40, 10 and 5) accuracy comparison of methods on the PDBBind splits for pocket level docking based on different metrics. The number k is specified in parenthesis.

| | Sequence Similarity Split | | | | | | Time Split | | | | | |
| | Distance-Pocket | | | Radius-Pocket | | | Distance-Pocket | | | Radius-Pocket | | |
| | %<1 | %<2 | Med. | %<1 | %<2 | Med. | %<1 | %<2 | Med. | %<1 | %<2 | Med. |
|---|---|---|---|---|---|---|---|---|---|---|---|---|
| HarmonicFlow (40) | 30.7 | 63.2 | 1.5 | 16.1 | 46.8 | 2.1 | 35.8 | 66.9 | 1.3 | 21.9 | 52.4 | 1.9 |
| IDFlow (40) | 34.3 | 64.0 | 1.4 | 18.6 | 48.0 | 2.1 | 37.6 | 67.1 | 1.2 | 27.8 | 53.4 | 1.8 |
| HarmonicFlow (10) | 22.0 | 53.9 | 1.8 | 9.1 | 36.7 | 2.4 | 25.8 | 58.8 | 1.6 | 15.8 | 45.4 | 2.2 |
| IDFlow (10) | 25.5 | 56.2 | 1.7 | 14.2 | 40.9 | 2.4 | 29.6 | 60.9 | 1.5 | 22.3 | 47.4 | 2.2 |
| HarmonicFlow (5) | 15.4 | 47.5 | 2.1 | 7.7 | 31.2 | 2.8 | 19.3 | 55.0 | 1.7 | 11.8 | 42.0 | 2.5 |
| IDFlow (5) | 20.6 | 52.0 | 1.9 | 10.0 | 34.3 | 2.6 | 24.7 | 57.9 | 1.6 | 17.8 | 44.9 | 2.4 |

**Energy reduction.** Fig. 5 presents L2-error energy minimization comparison during sampling for IDFlow and HarmonicFlow averaged over the test set of radius pocket time split. While the idempotency is also not perfect for IDFlow with the continuous loss minimization objective can never achieve an absolute idempotency, it's apparent that the energy converges much faster and better for IDFlow.

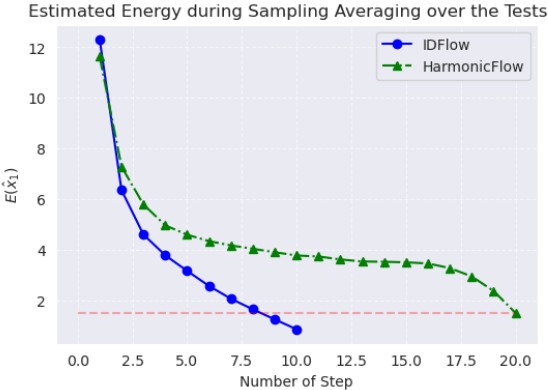

*Figure 5.* L2 error at each step during sampling averaging over the test set for HarmonicFlow and IDFlow. The energy of IDFlow is calculated by L2 error after refinement at each step, with total of 10 steps. For HarmonicFlow, we sample 20 steps with refinement for computing the L2 error, but use the output of the first iteration to simulate the ODE. The red dashed line marks the lowest energy of HarmonicFlow.

**Results on SCOPe.** In Table 7, we find that IDFlow can generate more designable proteins compared to the baseline FrameFlow and appears to be equally diverse and novel. Besides, the results over three runs are also more stable than the FrameFlow in terms of standard deviation. This demonstrates the method's effectiveness in generating highly designable proteins (relatively small, length < 128).

*Table 7.* Protein Backbone Generation on SCOPe. 10 backbones for the protein length from 60 to 128, in total 690 proteins. ∗ denotes the retrained FrameFlow. The results of FRAMEFLOW and GAFL are extracted from (Wagner et al., 2024) recomputed for mean and standard deviation.

| METHOD | DESIGNABILITY (↑) | DIVERSITY (↓) | NOVELTY (↓) |
|---|---|---|---|
| FRAMEFLOW | 81.2 | 0.37 | - |
| FRAMEFLOW* | 90.1 ± 1.7 | 0.39 ± 0.01 | 0.80 ± 0.01 |
| GAFL | 90.2 ± 0.6 | 0.38 ± 0.02 | - |
| **IDFLOW** | **90.9 ± 1.4** | **0.38 ± 0.02** | **0.78 ± 0.00** |

**Number of steps.** Fig. 6 analyzes the impact of NFEs on IDFlow and FrameFlow for the SCOPe dataset on both designability

and diversity. Both methods exhibit performance degradation when NFEs are reduced below 50. As shown in Fig. 6a, under identical sampling budgets, IDFlow achieves higher designability metrics and demonstrates greater robustness when sampling with fewer steps (30 NFEs). This suggests IDFlow's better stability in low-step regimes. Fig. 6b presents the diversity metrics versus the NFEs for IDFlow and FrameFlow. Again, similar to the designability, IDFlow appears to be more robust to the NFEs and generates relatively more diverse structures.

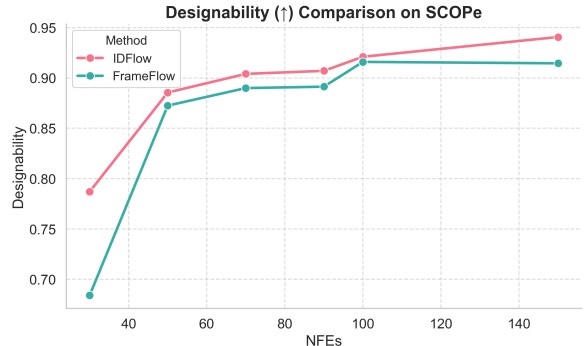
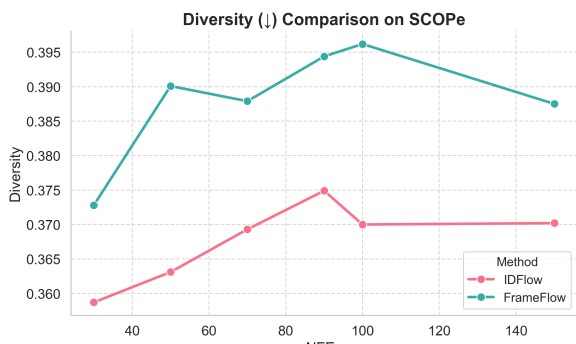

(a) Designability (↑) comparison between IDFlow and Frame-Flow on SCOPe dataset.

(b) Diversity (↓) comparison between IDFlow and FrameFlow on SCOPe dataset.

*Figure 6.* Designability and diversity comparison vs. NFEs on SCOPe dataset.

# G. Visualization

Figure 7 exhibits some randomly selected generated molecules compared with the Ground Truth. The sample is generated by 20 NFEs.

Figure 8 shows the designable backbones generated by IDFlow of various protein lengths. The sample is generated by 200 NFEs.

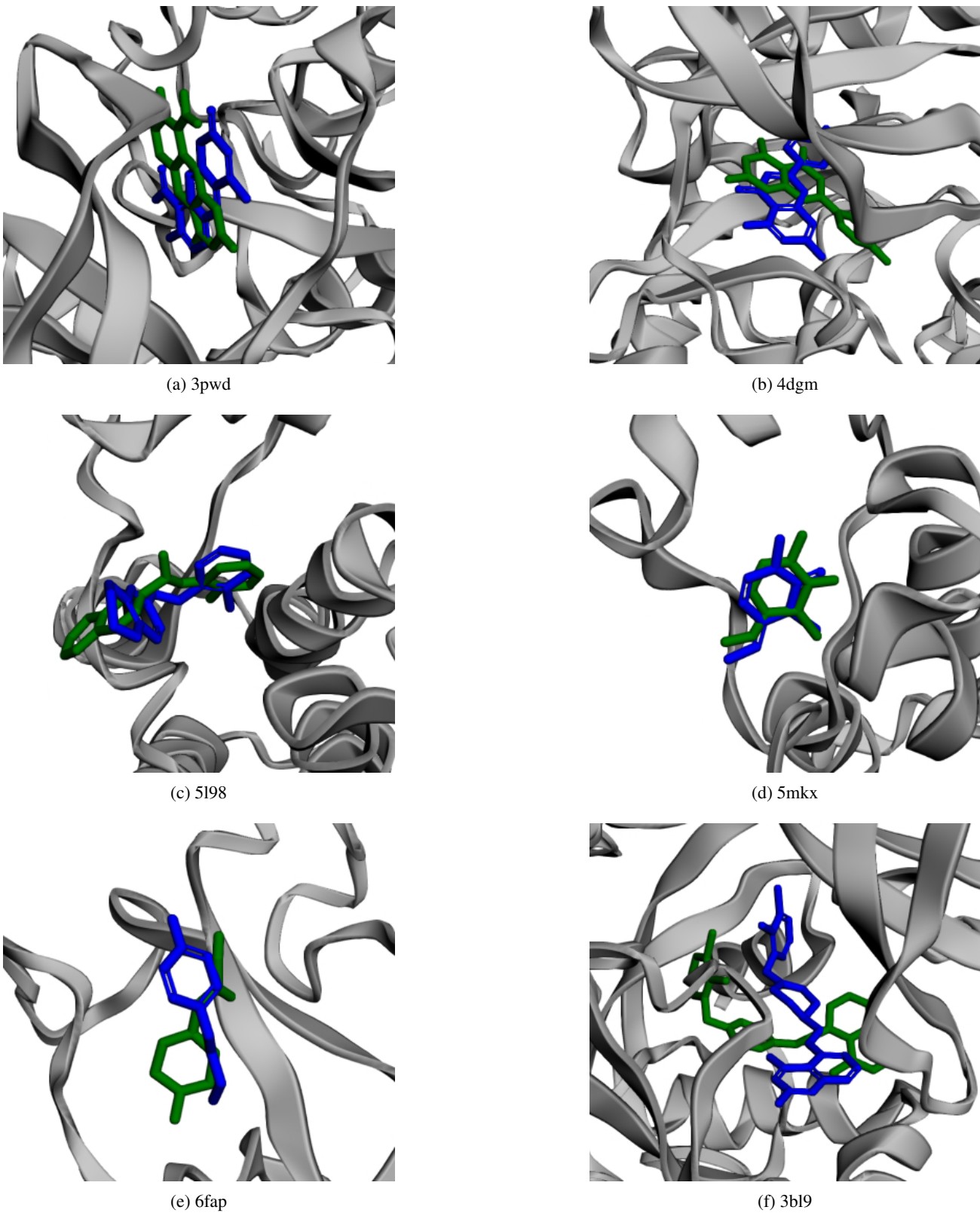

(a) 3pwd

(b) 4dgm

(c) 5l98

(d) 5mkx

(e) 6fap

(f) 3bl9

*Figure 7.* Six randomly selected generated complexes on the radius pocket docking on 30% sequence similarity split. The one in blue is the ground truth, and the one in green is generated from IDFlow.

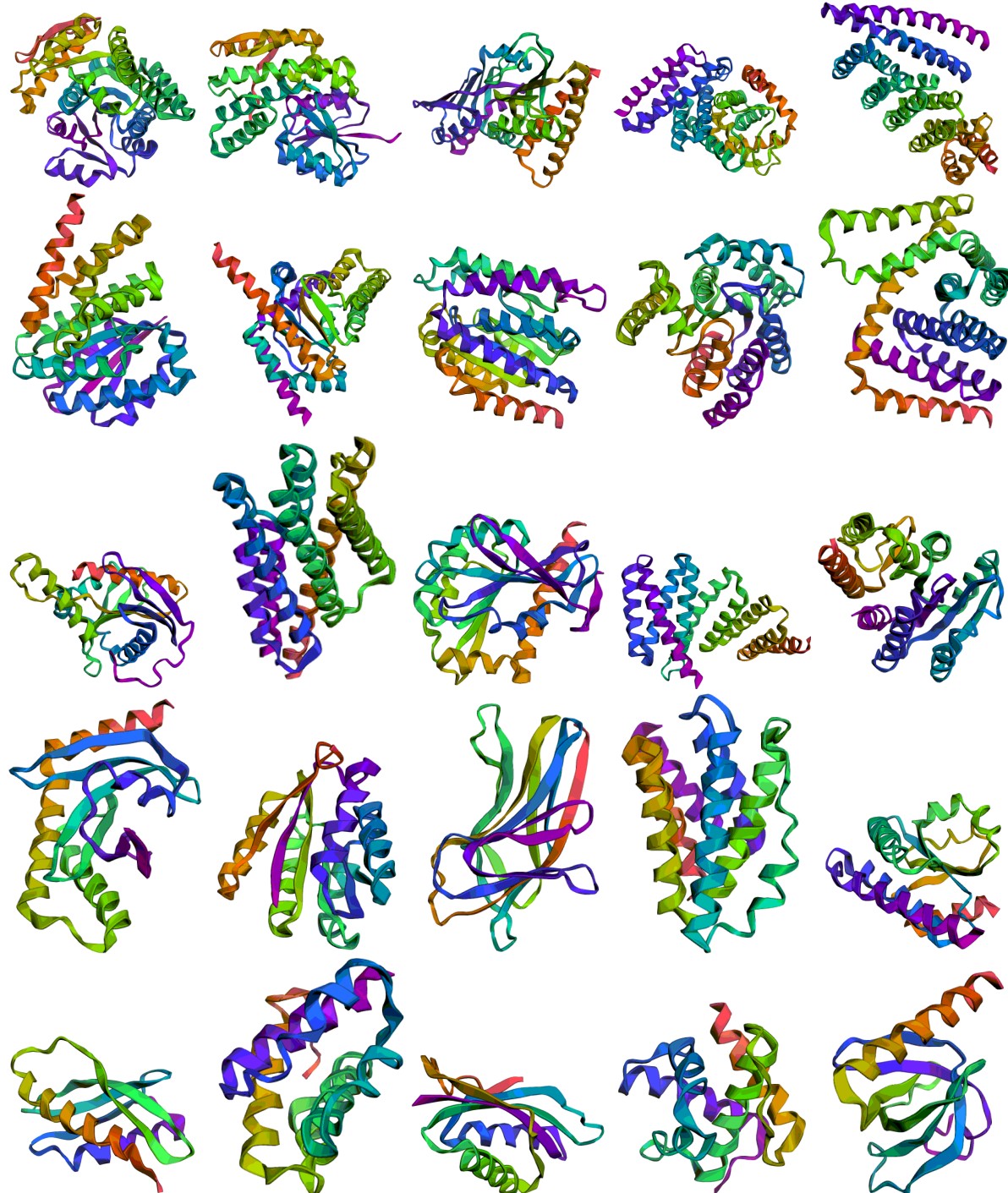

*Figure 8.* Designable Backbones at length [100, 150, 200, 250, 300] trained on PDB. Protein length from top to bottom [300, 250, 200, 150, 100].

