# OpenReview forum: "Energy-Based Flow Matching for Generating 3D Molecular Structure"
_ICML.cc/2025/Conference — ICML 2025 poster_

### Official Review · Reviewer_AZnb · 2025-02-23

**Overall Recommendation:** 3

**Summary:**

This paper proposes an enhanced flow matching framework, improving the standard setup from the energy-based perspective.
It provides a specific instance called IDFlow, which employs the reconstruction error as the energy function during training and then iteratively predicts and refines the sample.
Extensive experiments are conducted to validate the improvements of IDFlow over the standard setup, demonstrating its effectiveness.

## update after rebuttal
The authors have addressed some of my concerns.
However, I still think that comparing only one pair (one standard method with its corresponding IDFlow) for each task is insufficient to provide a comprehensive evaluation. After all, IDFlow is a modified flow matching that relies on the specific flow matching implementation, so it's necessary to validate its effectiveness and reliability in a broader context. Therefore, I decided to maintain my score but respect any final decision.

**Claims And Evidence:**

Some claims made in the submission are not supported by clear and convincing evidence:
* The specific relationship between energy-based models and the proposed method. For example, since the definition and goal of Eq.13 and Eq.18 are obviously different, why can the proposed method be interpreted from the energy-based perspective?
* The rationality of the idempotent flow map. Based on Eq.18, it's not enough to claim the flow map is idempotent.

**Essential References Not Discussed:**

No.

**Experimental Designs Or Analyses:**

The experimental designs and analyses are suitable for validating the effectiveness of the proposed method.

**Methods And Evaluation Criteria:**

The proposed method makes sense for the problem at hand.

**Other Comments Or Suggestions:**

1. Since IDFlow is a modified flow matching that relies on the specific flow matching implement, comparing only one pair (one standard method with its corresponding IDFlow) for each task is insufficient to provide a comprehensive evaluation. For example, it's necessary to report the performance of IDFlow based on various implemented versions of FoldFlow.
2. According to lines 323-327, the sampling steps for HarmonicFlow and IDFlow are the same. If so, it would be unfair to HarmonicFlow, as IDFlow involves two NFEs in a single step.

**Other Strengths And Weaknesses:**

No.

**Questions For Authors:**

1. The main contribution of this paper is just the incorporation of the refiner operation into flow matching, how to understand the relation between them and energy-based models?
2. For the sampling algorithm (Algorithm 2), how does the performance change when the refiner operation is removed? Additionally, how does the performance vary when the iteration count of the refiner operation is increased?  Moreover, why is $k$ a random integer between 1 and $K_{max}$ during training, but fixed to 1 during sampling? It's significant unmatched for the refiner operation between training and sampling.

**Relation To Broader Scientific Literature:**

This paper proposes a modification to the flow matching objective by directly minimizing an energy function, making it relevant to various applications of flow matching.

**Theoretical Claims:**

This paper claims that the standard setup of flow matching can be improved from the energy-based perspective, but does not provide the corresponding complete proof.

---

> ### Author Rebuttal · Authors · 2025-04-01
>
> Thank you for the effort to review our work! Here is our answer for addressing your concerns and a link to additional figures.
>
> https://anonymous.4open.science/r/ICML2025R-F85D/
>
> >Since the definition and goal of Eq.13 and Eq.18 are obviously different, why can the proposed method be interpreted from the energy-based perspective?
>
> First, we want to clarify that Eq. 18 is the proposed objective to learn the refiner G that maps the $\hat{x}_1$ onto its corresponding point $x_1$ on the data manifold. One way of viewing Eq. 18 from the energy-based point of view is if the G and f shares the same network (our case), the landscape is not only shaped by the trajectory sample $x_t$ (Eq. 7), but also the generated sample $\hat{x}_1$. This is associated with one way of training the EBMs, which maps the point off the manifold back to the data manifold [2]. Besides, the idempotency training also encourages the network to traverse over the L2 energy landscape to find a locally smooth solution, which can potentially lead to better generalization for being adversarial robust.
>
> >Based on Eq.18, it's not enough to claim the flow map is idempotent.
>
> We acknowledge the fact that empirically idempotency can not be guaranteed, as the continuous loss minimization objective could not be perfectly optimized unless a rigid structure is imposed. We do not claim that idempotency is achieved but only encouraged through the loss function. In the anonymous link (L2-Error-TestTime.png), we add an L2 error reduction plot during sampling for HarmonicFlow and IDFlow averaged over the time split test set (363 examples). This shows that even if the absolute idempotency is not achieved (0 across the timestep), the IDFlow yields better idempotency that the L2 error converges faster and better than the baseline.
>
> > it's necessary to report the performance of IDFlow based on various implemented versions of FoldFlow.
>
> We acknowledge that the additional experiments on FoldFlow could further strengthen the claim. However, FrameFlow is conceptually very similar to the core idea of FoldFlow-OT and FoldFlow-Base, which both apply the SE(3) flow matching to generate the protein backbones. The only deviation is the FoldFlow-SFM introducing the stochastness for sampling using the IGSO(3) distribution. Considering the high similarity of the methods, we would expect a similar performance improvement over FoldFlow.
>
> > According to lines 323-327, the sampling steps for HarmonicFlow and IDFlow are the same. If so, it would be unfair to HarmonicFlow, as IDFlow involves two NFEs in a single step.
>
> We want to clarify that our comparison is fair. HarmonicFlow adopts 20 steps (equivalent to 20 NFEs) for sampling, specified in Appendix E-“Hyperparameter” [1], while IDFlow uses 10 steps (also equivalent to 20 NFEs). We apologize for the confusion and will clarify in the next version.
>
> >How to understand the relation between refiner and energy-based models?
>
> We could think of the refiner as a means to map the generated sample to be minimal of a certain energy function. The energy function is imposed by defining a loss function on the refiner output. In the paper, we use L2-error because of its simplicity and alignment with performance metrics (RMSD) and the nice properties of idempotency. In this case, this refiner maps the point off the data manifold \hat{x}_1 to the data manifold (similar to the denosing autoencoder), a contrastive approach for training the EBMs [2]. Generally speaking, we can impose different energy losses, such as the flat bottom potential in [3], on the refiner output to improve the chemical plausibility of the generated sample. Besides, it can also integrate with the pretrained force field that, instead of refining the sample to the minimum, one can also leverage the Langevin diffusion to sample from the distribution governed by the energy function. A detailed description can be found in the response to the reviewer rcEp in paragraph 3.
>
> >It's significant unmatched for the refiner operation between training and sampling.
>
> Refining the sample multiple times at each step increases the sampling budget. In the link (TestTime-K-Ablation.png), we ablate k while keeping the total NFEs constant (20 or 21), using more discretizations for smaller k. Ideally, an idempotent function wouldn’t require refinement at test time, but empirical results show that setting k=1 yields the best performance. We attribute this to large discretization errors for larger k, as more NFEs are needed for idempotency. Setting k=1strikes the best tradeoff between idempotency and discretization.
>
> [1] Harmonic Self-Conditioned Flow Matching for joint Multi-Ligand Docking and Binding Site Design, ICML 2024.
>
> [2] LeCun, Y. From machine learning to autonomous intelligence: Lecture 2, 2022. URL https://leshouches2022.github.io/SLIDES/lecun-20220720-leshouches-02.pdf.
>
> [3] Composing Unbalanced Flows for Flexible Docking and Relaxation, ICLR 2025.

---

> > ### Comment · Reviewer_AZnb · 2025-04-09
> >
> > Thanks for your response, which partially addresses my concerns.
> >
> > However, I still think that comparing only one pair (one standard method with its corresponding IDFlow) for each task is insufficient to provide a comprehensive evaluation. After all, IDFlow is a modified flow matching that relies on the specific flow matching implementation, so it's necessary to validate the effectiveness and reliability of IDFlow in a broader context.
> > I will keep my score but respect any final decision.

---

### Official Review · Reviewer_rcEp · 2025-03-12

**Overall Recommendation:** 4

**Summary:**

The authors propose an energy-based perspective of flow matching for the purpose of improving the quality of 3D structures predicted by generative models. This perspective leads to a modified training procedure called Idempotent Flow Map Training, which trains a network to produce predictions by iterative refinement, where the initial output of the network is again passed into the network. In particular, the network uses the “denoiser” or “x” parameterization. This simple modification enables consistently better prediction quality across experiments in docking and protein backbone generation, with no increase in inference cost, but at an increased training cost.

**Claims And Evidence:**

The claim of an energy-based perspective of flow matching is a bit strained since the paper only ever uses L2 reconstruction error as the energy function. At least one experiment using another energy function should be shown to support this claim. This would also help support the claim that the proposed method is designed for structure generation models. As far as I can tell, the proposed method is applicable to any diffusion/flow model which can use the “denoiser” parameterization, and so this method would be applicable for images and other modalities as well.

The proposed method does not seem to rely on the energy-based perspective. The ablation of $K_max$ in Table 4 indicates that setting $K_{max}$ greater than 0 is what is responsible for improved performance. However, Idempotent Flow Map training could just as easily be thought of as introducing a bit of simulation-during-training to the standard flow matching training procedure, and providing extra training supervision to each step of simulation-during-training.

The claim that IDFlow trains faster is uncertain to me. While the experimental comparison uses the same architecture and same number of epochs for training, IDFlow uses more compute per forward pass due to the requirement of simulation-during-training in the refinement loss. Figure 4 in Appendix F demonstrates that IDFlow trains with fewer epochs, but it would be valuable to see what these curves look like if the x-axis were switched to wall-clock time. This is important since it appears that both models are undertrained (validation metrics have not completely plateaued yet). It would also be an interesting baseline to compare to simply increasing the size of the model architecture.

**Essential References Not Discussed:**

The focus on idempotency is very similar to Consistency Models [1], although Consistency Models focus on reducing the number of generation steps rather than increasing the quality of generated samples. Idempotency Flow Map Training can be seen as an instance of Consistency Training except that the training target sometimes comes from multiple steps of simulation during training.

Consistency models are only idempotent at t=0, where the skip connection forces the network to be the identity function. In contrast, IDFlow is idempotent for every t.

However, the sampling approach is different from Consistency Models.


[1] Song, Y., Dhariwal, P., Chen, M., & Sutskever, I. (2023). Consistency models.

**Experimental Designs Or Analyses:**

I checked that the experiments on molecular docking and protein backbone generation closely follow experiments executed in previous work. The only concern is how much the training cost increases by.

**Methods And Evaluation Criteria:**

In the area of molecular structure prediction, the proposed experiments are complete and make sense. However, the proposed method appears to be much more general than the application area of 3D structure prediction, and I am curious how the proposed training procedure would affect an image model.

**Other Comments Or Suggestions:**

Typos:
line 641: "seminar"
line 240, left column: “L_CFM” should be “L_G”
line 947: “We train the model 8 on” should be “We train the model on 8”

use of $K_{max}$ vs $k_{max}$ is not consistent

3D structure prediction is relevant to many more areas of chemistry than just biomolecules. For example, it is relevant for moment-constrained structure elucidation [1] and crystal structure prediction [2] [3]. Crystal structure prediction in particular has traditionally focused on finding the lowest-energy structures, where energy is given by DFT.

[1] Cheng, A., Lo, A., Lee, K. L. K., Miret, S., & Aspuru-Guzik, A. (2024). Stiefel Flow Matching for Moment-Constrained Structure Elucidation. arXiv preprint arXiv:2412.12540.
[2] Jiao, R., Huang, W., Lin, P., Han, J., Chen, P., Lu, Y., & Liu, Y. (2023). Crystal structure prediction by joint equivariant diffusion. Advances in Neural Information Processing Systems, 36, 17464-17497.
[3] Zeni, C., Pinsler, R., Zügner, D., Fowler, A., Horton, M., Fu, X., ... & Xie, T. (2023). Mattergen: a generative model for inorganic materials design. arXiv preprint arXiv:2312.03687.

**Other Strengths And Weaknesses:**

The paper provides thorough background knowledge on flow matching and experimental setup with molecular structure prediction.

The notation is sometimes confusing, with networks like $G_\theta$ and $E_\theta$ defined for the sake of abstracting out an energy function, only to always set the energy equal to the L2 loss.

Simple and effective methods are valuable, but appear less novel when it is not as clear why these simple changes provide incremental improvements. If I were to critique the method, it appears that the proposed method simply adds simulation-during-training and provides extra training supervision by calculating loss

**Questions For Authors:**

Can you include Figure 4 but with wall-clock time as the x-axis? This would help clearly demonstrate the effectiveness of IDFlow.
If IDFlow has a higher wall-clock training time than original baselines, can you compare IDFlow to the performance of a larger model with similar memory and training throughput?
Can you provide at least one experiment of how the framework of energy-based flow matching would work for an energy function that is not the L2 loss? For example, alanine dipeptide with either a classical force field or xTB, or use a neural force field such as https://fair-chem.github.io/core/quickstart.html
(less priority) Can you demonstrate a simple application to images, such as CIFAR-10?

**Relation To Broader Scientific Literature:**

The paper provides a simple method for improving generation quality for flow/diffusion models, which is relevant to any work that applies these generative models.

Specifically, the key contribution of the paper is Idempotent Flow Map training, which is related to recycling/self-conditioning. This paper demonstrates that these tricks empirically improve performance, though I am still unsure of the full reason except that model expressivity is increased.

The concept of learning an idempotent map is very related to consistency models (see below).

**Theoretical Claims:**

I checked the derivations in the appendix and found no issues.

---

> ### Author Rebuttal · Authors · 2025-04-01
>
> Thank you for the effort to review our work! Here is our answer for addressing your concerns and a link to additional figures.
>
> https://anonymous.4open.science/r/ICML2025R-F85D/
>
> >Can you include Figure 4 but with wall-clock time as the x-axis?...
>
> Thanks for pointing out the increased throughput of IDFlow. In the link we provide a figure of the validation metric  (ValidationMetrics.png) with the x-axis to be wall clock time. The HarmonicFlow-L model has 60 scalar and 10 vector features for the tensor field network, totaling 16.3M parameters, compared to 5.7M in HarmonicFlow and IDFlow. The figure shows the better validation performance with a larger model size, but still lower than the IDFlow. Following the reviewer’s advice, we compare the HarmonicFlow-L with IDFlow in the link (TimeSplitRadiusDocking.png), which demonstrates that idempotency does not simply come from the increased training throughput. It also shows the improvement over baseline with fewer model parameters.
>
> >Can you provide at least one experiment of how the framework of energy-based flow matching would work for an energy function that is not the L2 loss? For example, alanine dipeptide with either a classical force field or xTB, or use a neural force field such as https://fair-chem.github.io/core/quickstart.html (less priority)
>
> We thank the reviewer for suggesting the use of pre-trained forcefields or DFT energies in our framework! We describe how to incorporate this approach: the pre-trained forcefield can be directly integrated into the sampling algorithm, with the Langevin diffusion process refining the sample as follows: $\hat{x}_1 = \hat{x}_1 - \frac{\epsilon^2}{2} \nabla E(\hat{x}_1) + \epsilon z$ where $\epsilon$ is the step size and $z$ is a sample from a standard Gaussian. The gradient can be computed using the pre-trained forcefield, and the energy can be derived from DFT. We will include this in the next version of the paper. However, introducing a pre-trained forcefield may unfairly advantage the baselines due to the added complexity and capacity. The core innovation of our approach is using a refiner $G$ parameterized by the same network to refine samples toward the minimum of the energy function with minimal training overhead. The energy function is defined by the loss on the refiner's output, and the idempotent flow map arises from using the same loss function as the CFM loss.
>
> In the link, we also present additional experimental results (TimeSplitRadiusDocking.png) on time-split radius pocket docking, where we use a different energy function more aligned with the molecular setup. The new energy function (EB-FM) consists of three terms: 1) the reconstruction errors 2) the ligand bond distance as intramolecular potential 3) the protein ligand distance as intermolecular potential. Without further tuning the weights of different energy losses, the EB-FM achieves better performance on the RMSD median. We will include the full results in the next version.
>
> > Can you demonstrate a simple application to images, such as CIFAR-10?
>
> We thank the reviewer for raising the potential application to image data. Idempotency is a general idea that can be applied to many different types of generative models and different modalities. Considering higher dimensionality and the weaker theoretical connection of images to the stability and energy assumption, we'd like to leave this to future work.
>
> >The concept of learning an idempotent map is very related to consistency models.
>
> We mildly disagree with the opinion that IDFlow is an instance of consistency models (CM). Key distinctions are as follows. First, consistency training minimizes the discrepancy between noisy data at neighboring steps, while flow matching embeds consistency by outputting clean samples across noise levels. In this sense, CM (PF-ODE) is closer to standard flow matching data parameterization. Consistency training enables fast sampling, while flow matching still requires ODE simulation. Second, an idempotent flow should ideally have zero velocity at $𝑡=1$ whereas CM has a non-zero, unstable vector field at $t=1$. Idempotency can also be applied to CM, making the learned consistency function also idempotent.
>
> > Idempotent Flow Map training could just as easily be thought of as introducing a bit of simulation-during-training....
>
> From idempotent inference perspective, refining $\hat{x}_1$ during training adds some simulation, whereas vanilla flow matching relies on ODE simulation without idempotent inference. Introducing idempotency into flow matching ties into key concepts like physical stability in molecular dynamics. Training with idempotent flow reduces network uncertainty by incorporating the domain knowledge that generated molecules should be stable.
>
> Lastly, we thank the reviewer for highlighting the broader impact of 3D structure generation and for pointing out typos. We will address these in the next version by including more references and improving the writing.

---

### Official Review · Reviewer_Z5a6 · 2025-03-13

**Overall Recommendation:** 4

**Summary:**

This paper introduces a new method to train flow matching models. They want to sample from an energy function and this comes with very interesting outcomes. By using an energy function based on a squared euclidean distance, they realize that it boils down to train an indepotent map. To make the training efficient, they follow the self-conditioning procedure where they train the indepotent map 50% of the time and use regular flow matching 50% of the time. The sampling also uses only two function evaluation to be compute efficient. They evaluated their method on protein backbone generation and molecular docking where it outperforms the methods it is built on.

**Claims And Evidence:**

They claim to define a flow matching method that sample form a distribution and this claim is true. They claim to achieve better results than the existing flow matching method and this is also true.

**Essential References Not Discussed:**

NA

**Experimental Designs Or Analyses:**

THey follow the literature. The authors did a sensitivity analysis with respect to the number of function evaluation.

**Methods And Evaluation Criteria:**

The methods is very relevant and interesting. The evaluation follows the standard practice in the literature.

**Other Comments Or Suggestions:**

See above.

**Other Strengths And Weaknesses:**

The paper is not always well written. I got a little lost in the paragraph Energy relaxation, confidence model and the EBMs. Maybe the author can rewrite it.

**Questions For Authors:**

Can we couple it with self-conditioning?

**Relation To Broader Scientific Literature:**

The literature review is complete.

**Theoretical Claims:**

Not applicable

---

> ### Author Rebuttal · Authors · 2025-03-31
>
> Thank you for the effort to review our work! Here is our answer for addressing your concerns.
>
> >The paper is not always well written. I got a little lost in the paragraph Energy relaxation, confidence model and the EBMs. Maybe the author can rewrite it.
>
> Thanks for pointing out the writing about the paper, specifically the section 3.2. We would make the writing clearer in the next version. In section 3.1, we want to connect the structure relaxation techniques and the confidence models to the EBM concept that the generated samples $\hat{x}_1$ should associate with the minimum of the energy function. This energy function could be the physical energy function, like the one used in structural relaxation, or the confidence model output used for ranking the sample.
>
> >Can we couple it with self-conditioning?
>
> We thank the reviewer for bringing the self-conditioning into discussion. We want to stress that the idempotency is orthogonal to the self-conditioning. Self-conditioning is still doing input conditioning where the network is conditioned by the $\hat{x}_1$ to predict the data $x_1$ (in our setting) $\textbf{given the input $x_t$}$. The conditioning is usually achieved by concatenation for images or as edge features for molecules. In our case, since the self-conditional information is injected through the edge features, it can be nicely integrated with the IDFlow. The implementation requires a double “$50\\%$”: for each training step, we first decide if to activate the self-conditioning and then idempotency training. This results in $25\\%$ time self-conditioning FM training, $25\\%$ time non-self-conditioning FM training, $25\\%$ time self-conditioning idempotency training, and $25\\%$ time non-self-conditioning idempotency training.

---

### Official Review · Reviewer_Fdeb · 2025-03-14

**Overall Recommendation:** 3

**Summary:**

This paper proposes to enhance the flow-matching framework with an energy-based perspective to learn iterative mapping. Such an idempotent mapping, as demonstrated theoretically in the paper, has better stability during generation. Experiments on protein docking and generation demonstrate better generation quality.

**Claims And Evidence:**

The authors proposed an alternative flow-matching objective inspired by energy-based models. In their experiments, the proposed model achieved superior performance over the baselines. A few more ablation studies (see "Experimental Design" section) would be more convincing.

**Essential References Not Discussed:**

I believe essential references have been discussed in this paper.

**Experimental Designs Or Analyses:**

While most of the presented results can demonstrate the superior performance of the proposed approach, the following aspects can be better verified (theoretically or empirically) for more convincing claims.

- Algorithm 1 adopted the combination of the flow matching loss ($L_G$ in Eq.18) and the idempotent loss ($L_R$ in Eq.21). Intuitively, the ratio should be important, as the zero mapping $f_{\theta,t}(x)\equiv0$ trivially satisfy the idempotency condition but does not give the correct clean data. The authors should also demonstrate the impact of the probability $m$ for balancing between these two losses.
- Intuitively, if the learned denoiser is perfectly idempotent, one-step (or few-step) generation can be performed as the model will generate a consistent prediction. However, it seems that in Figure 3, the proposed model still requires multiple NFEs to achieve descent. This seems to indicate that the learned model is not close to idempotent. The authors should verify this.

**Methods And Evaluation Criteria:**

The method in the paper is well-supported, with both good intuitions and theoretical results inspired by energy-based models. However, some evaluation metrics on unconditional protein generation were questionable.

- It is unclear why the authors separated the models and the baselines into two categories in Table 3, and only compared models within the category.
- For multiple baselines with the same better performance score, only one was highlighted in bold, which may be misleading.
- For the comparison of time, the authors tried to unfairly compare models with different numbers of sampling steps.

**Other Comments Or Suggestions:**

Some notations can be improved to be more consistent. For example, the authors used different fonts in Algorithm 1 for the same variable.

**Other Strengths And Weaknesses:**

In the unconditional protein generation task, the authors implicitly applied the proposed framework to Riemannian manifolds, which could be substantially different. For example, the equivalence of the target prediction versus the vector field prediction only holds for the Euclidean manifold (or zero-curvature manifolds) but would fail for general manifolds. Specifically, on SO(3), **the target prediction in Eq.36 differs from the Riemannian flow matching loss in Eq.35**. The authors should explicitly note this difference, which does not lead to Riemannian flow matching or any of its theoretical benefits. In this way, it might be better to formulate the proposed approach as a standalone generative framework instead of a variant of flow matching.

**Questions For Authors:**

See other sections for questions.

**Relation To Broader Scientific Literature:**

This work has a potentially broader impact on scientific domains including protein design, protein docking, and other generative tasks in AI4Science domains. I suggest the author also discuss such applications in downstream tasks.

**Theoretical Claims:**

The theoretical grounds of energy-based models are clearly stated and discussed in this work.

---

> ### Author Rebuttal · Authors · 2025-03-31
>
> Thank you for the effort to review our work! Here is our answer for addressing your concerns.
>
> >For the comparison of time, the authors tried to unfairly compare models with different numbers of sampling steps.
>
> We want to clarify that our comparison is fair. HarmonicFlow adopts 20 steps (equivalent to 20 NFEs) for sampling, specified in Appendix E-“Hyperparameter” [1], while our method uses 10 steps (also equivalent to 20 NFEs). We apologize for the confusion and will clarify in the next version.
>
> > Intuitively, the ratio should be important, as the zero mapping $f_{\theta,t}(x)=0$ trivially satisfy the idempotency condition but does not give the correct clean data. The authors should also demonstrate the impact of the probability $m$ for balancing between these two losses.
>
> We appreciate the point raised by the reviewer that idempotency training can lead to a trivial solution. To resolve this, we adopt a simple approach where during training network input is detached from the computation graph (Eq. 21) for $L_R$, so that there is no information leak from previous iterations affecting the training dynamics. With our 50% training setup, the two losses are equally balanced with m from uniform distribution. This strategy aligns with the sampling algorithms, which at each step we first predict the clean sample and then refine it.
>
> >It seems that in Figure 3, the proposed model still requires multiple NFEs to achieve descent. This seems to indicate that the learned model is not close to idempotent. The authors should verify this.
>
> We acknowledge the fact that empirically idempotency can not be guaranteed, as the continuous loss minimization objective could not be perfectly optimized unless a rigid structure is imposed. We do not claim the idempotency is achieved but only encouraged through the loss function. In the link (https://anonymous.4open.science/r/ICML2025R-F85D/L2-Error-TestTime.png), we provide an L2 error reduction plot during sampling for HarmonicFlow and IDFlow averaged over the time split test set. This shows that even if the absolute idempotency is not achieved, the IDFlow yields better idempotency.
>
> >This work has a potentially broader impact on scientific domains including protein design, protein docking, and other generative tasks in AI4Science domains. I suggest the author also discuss such applications in downstream tasks.
>
> Here are some discussions. First, the proposed energy-based framework can potentially improve the chemical plausibility of generated molecules without too much training overhead. The refiner can be purposed to refine the sample to a distribution governed by a certain energy function that the practitioners are interested in. It can also be integrated with pre-trained forcefield. More details can be found in paragraph 4 of the rebuttal to the reviewer rcEp. These relate to some other AI4Science problems such as crystal structure or molecular structure elucidation in chemistry. Second, the idea of idempotency also has a potential impact as the idempotency encourages the network to traverse over the loss landscape to find a locally smooth solution, which can potentially lead to better generalization of generative models for being adversarial robust.
>
> > Specifically, on SO(3), the target prediction in Eq.36 differs from the Riemannian flow matching loss in Eq.35.
>
> We thank the reviewer for raising a valid point regarding SO(3) parameterization. While Equations 35 (Euclidean) and 36 (manifold) parameterize flows differently, both leverage geodesics constructing the flow path---key to Riemannian flow matching’s theoretical strength. The rotation field can be computed as: $\frac{\log_{r_t}(\hat{r}_1)}{1-t}$ ensures the predicted rotation $\hat{r}_1$ aligned with SO(3)’s geometry to follow the geodesics. However, since rotation is parameterized through the quaternions, and quaternions’ double cover of $\mathrm{SO}(3)$ introduces non-uniqueness, which may potentially increase the learning difficulty of the network.
>
> >Some notations can be improved to be more consistent. For example, the authors used different fonts in Algorithm 1 for the same variable. For multiple baselines with the same better performance score, only one was highlighted in bold, which may be misleading.
>
> Thanks for highlighting the notation inconsistencies. We will revise Table 3 to separate baselines, clarify improvements over FrameFlow, boldface all state-of-the-art results, and correct all notation/writing inconsistencies in the next version.
>
> [1] Harmonic Self-Conditioned Flow Matching for joint Multi-Ligand Docking and Binding Site Design, ICML 2024.

---

### Decision · Program_Chairs · 2025-05-01

**Decision:**

Accept (poster)

**Comment:**

This paper presents an energy-based formulation of flow matching for 3D molecular structure generation. The authors instantiate their framework through IDFlow, which leverages reconstruction error as the energy function to better shape the loss landscape using contrastive samples generated by the flow model. IDFlow constructs a flow map that iteratively predicts and refines the sampling trajectory toward the target molecular structure. Experiments on protein binding and protein backbone generation are conducted to verify the effectiveness of the proposed model.

The paper received four reviews. Following the rebuttal and discussion, a consensus was achieved among all four reviewers, culminating in the decision to accept the paper. This agreement is attributed to the paper's innovation, promising results, and potentially broader impacts. The AC agrees with the reviewers and recommends accepting the paper. To further improve the paper quality, the AC urges the authors to revise their paper by taking into account all the suggestions provided by the reviewers, including integrating extra experimental results conducted during the rebuttal phase.